# Spatial-Temporal Distribution of the Freeze–Thaw Cycle of the Largest Lake (Qinghai Lake) in China Based on Machine Learning and MODIS from 2000 to 2020

Weixiao Han [1,2], Chunlin Huang [1], Juan Gu [3], Jinliang Hou [1,*] and Ying Zhang [1]

1 Key Laboratory of Remote Sensing of Gansu Province, Heihe Remote Sensing Experimental Research Station, Northwest Institute of Eco-Environment and Resources, Chinese Academy of Sciences, Lanzhou 730000, China; weixiaohan@lzb.ac.cn (W.H.); huangcl@lzb.ac.cn (C.H.); zhang_y@lzb.ac.cn (Y.Z.)

2 University of Chinese Academy of Sciences, Beijing 100049, China

3 Key Laboratory of Western China's Environmental Systems, Ministry of Education, Lanzhou University, Lanzhou 730000, China; gujuan@lzu.edu.cn

* Correspondence: jlhours@lzb.ac.cn

**Abstract:** The lake ice phenology variations are vital for the land–surface–water cycle. Qinghai Lake is experiencing amplified warming under climate change. Based on the MODIS imagery, the spatio-temporal dynamics of the ice phenology of Qinghai Lake were analyzed using machine learning during the 2000/2001 to 2019/2020 ice season, and cloud gap-filling procedures were applied to reconstruct the result. The results showed that the overall accuracy of the water–ice classification by random forest and cloud gap-filling procedures was 98.36% and 92.56%, respectively. The annual spatial distribution of the freeze-up and break-up dates ranged primarily from DOY 330 to 397 and from DOY 70 to 116. Meanwhile, the decrease rates of freeze-up duration (DFU), full ice cover duration (DFI), and ice cover duration (DI) were 0.37, 0.34, and 0.13 days/yr., respectively, and the duration was shortened by 7.4, 6.8, and 2.6 days over the past 20 years. The increased rate of break-up duration (DBU) was 0.58 days/yr. and the duration was lengthened by 11.6 days. Furthermore, the increase in temperature resulted in an increase in precipitation after two years; the increase in precipitation resulted in the increase in DBU and decrease in DFU in corresponding years, and decreased DI and DFI after one year.

**Keywords:** machine learning; Qinghai Lake; MODIS; Google Earth Engine; ice phenology





## 1. Introduction

Spatially explicit knowledge of lake ice is crucial for understanding a wide variety of earth system processes and interactions with the environment, including hydrological budgets, sediment trapping; heat fluxes and coupled weather and climate effects; lake productivity; species richness; food chain dynamics; and inland fishery yields [1,2]. The lake surface water–ice state is controlled by the local geolocation and climate conditions, especially the freeze-up dates and the break-up dates. The lake surface water's physical properties changed due to the transition of the lake surface water–ice state processes [3].

Known as the "Third Pole", the Tibetan Plateau (TP) is the highest and most extensive highland in the world. As the Asian water tower, there exist around 1200 lakes (>1 km$^2$) with an overall surface water area of 47,000 km$^2$, which makes up beyond 50% of the overall surface water area of Chinese lakes, including the largest lake: Qinghai Lake (4254.90 km$^2$) [4]. The Qinghai Lake locates in the Yellow River sub-basin, which is an area of approximately $2.5 \times 10^5$ km$^2$; the distinctive climatic conditions and geophysical environment make it a highly attractive research region [5].

Ice cover dominates the annual cycle of the lake and essentially isolates the water body from the exchange of moisture and gas fluxes [6]. Thus, lake ice cover plays a key role in regional water and energy balance [7]. Lake phenology events, such as critical

lake surface water freeze-up dates, lake ice break-up dates, ice cover duration, and land surface temperature (LST), could represent sensitive indicators for monitoring land–surface–water variation [8]. The frozen duration of Qinghai Lake was monitored using satellite passive microwave remote sensing low frequency data [9]. The lake ice phenology of Nam Co (Central Tibetan Plateau) derived from multiple MODIS data products was interpreted [10]. The role of climate and lake size in regulating boreal lakes' ice phenology was analyzed [11]. The lake ice phenology was extracted using the convolutional neural network method [12]. The lake ice phenology was mapped using MODIS/Terra L1B TOA (MOD02) product based on four machine learning classifiers (multinomial logistic regression, MLR; support vector machine, SVM; random forest, RF; gradient boosting trees, GBT) [13]. The sentinel-2 and auxiliary TanDEM-X topographic data were trained for automated mapping of Antarctic supraglacial lakes using RF [14]. Besides, impacts on lake ice cover and formation could impact on socio-economic elements such as ice roads, transportation, cultural recreation, and tourism [15]. The acquisition of lake surface water and lake ice cover data depends on classic methods, including in situ observations and aerial surveys. Nevertheless, these acquisition types are impacted by the local geophysical environment and untraversed sites. Because of the sparse collection of in situ data, complete long-term metadata are difficult to obtain [16]. Remote sensing data could atone for classic observation defects with various geospatial and temporal resolutions and be remarkably congruous with in situ observations. There are advantages in terms of high temporal and spatial resolutions in remote sensing data; as well as rich multispectral information, it can be used to investigate spatially distributed hydrological states for use in modeling, assessment, and management. However, cloud cover can obscure significant portions of the images in the visual wavelengths [17], therefore, the cloud gap-filling methods were used in the optical images [18].

In recent years, machine learning has been adopted for remote sensing applications, including neural networks, SVM, and RF [19,20]. RF was relatively insensitive to the choice of the hyperparameters compared to the other classifiers [13,21]. The RF algorithm is being increasingly applied to satellite and aerial image classification and continuous field datasets. It has the potential to simultaneously extract the daily surface water and ice cover of lakes, which are based on remote sensing data in the Google Earth Engine (GEE) platform. The latency of image availability is 21 days.

The previous research compared the water index, ice threshold method, and RF algorithm; the annual lake ice phenology dates were extracted during 2000–2018 based on MODIS in the GEE platform [22]. This work is a continuation of the previous research. It focuses on the spatial-temporal distribution of the ice phenology of the Qinghai Lake based on MODIS images (2000–2020), the Joint Research Center's Global Surface Water (JRC GSW), and the ERA5 dataset using the RF algorithm in the GEE platform. Additionally, the cloud gap-filling method was applied to reconstruct the gap area. The innovation is the construction of the lake information under the cloud cover pixels, and the changes in the lake ice phenology were refined from the whole lake to the pixel level. Section 2 describes the study area and dataset. Section 3 presents the methods, including RF classification, cloud gap-filling, evaluation approaches and definition of the lake phenology. Section 4 illustrates the results of cloud removal, the classification of RF, validation, and cloud gap-filling, and discusses the annual temporal-spatial distribution of freeze-up and break-up dates and temporal dynamics.

## 2. Data

### 2.1. Study Area

Qinghai Lake is a brackish endorheic lake located in the Yellow River sub-basin in the northeastern TP [23,24] (Figure 1). Qinghai Lake (36°32′–37°15′ N, 99°36′–100°47′ E) has a surface water area of 4500 km$^2$; the average lake depth is 21.0 m, the maximum depth is 32.8 m, the water volume of $71.6 \times 10^9$ m$^3$, and it has an elevation of 3194 m [25]. The lake is currently fed by several rivers, with a mean annual discharge of $1.69 \times 10^9$ m$^3$ [26].

Over the past 57 years, the annual average temperature was 1.9 °C [27]. The mean January and July temperatures of the recent 40 years are −11.4 °C and 12.5 °C, respectively [28]. Qinghai Lake accesses the surface water–ice freeze–thaw cycle period from November to April of the next year [29].

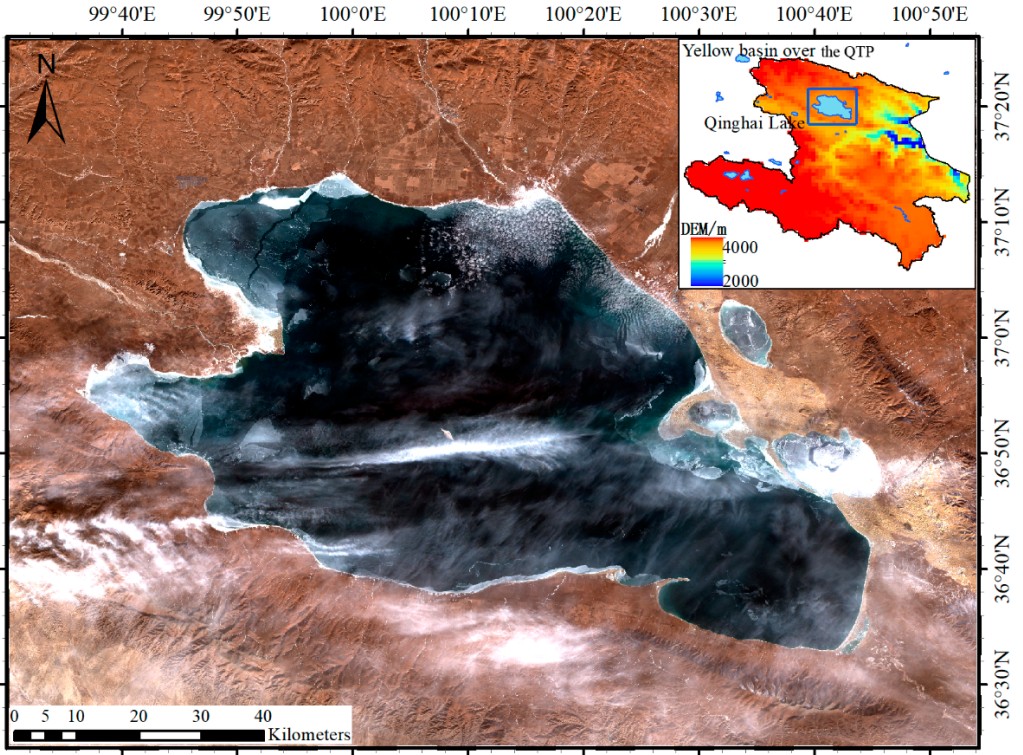

**Figure 1.** The geographical location of Qinghai Lake (Sentinel-2 image on 22 December 2017) in the upper reaches of the Yellow River Basin cross cover the TP [24]. (Note: This figure is created using Sentinel-2, and the source of the Yellow River Basin: https://data.tpdc.ac.cn/zh-hans/data/dff6b4 37-90a1-4729-8140-faafc544860f/ (accessed on 31 July 2020)).

### 2.2. MODIS Surface Reflectance

In order to extract the daily pixel wise spatial-temporal distribution of the water and ice, the daily MODIS surface reflectance was used. It includes seven-bands, which are bands 1–7 (1–2: 250 m, 3–7: 500 m resolution). It is provided by the MODIS Land Surface Reflectance Science Computing Facility [30]. The products are the Level 2 data derived from the Level 1B data, the radiometric calibration and the atmospheric correction being processed in Level 1 and Level 2, respectively. Therefore, the atmospheric scattering and absorption are eliminated. The MOD09GQ V6 and MYD09GQ V6 provided bands 1–2 surface reflectance, as well as the MOD09GA V6 and MYD09GA V6 provided bands 3–7 surface reflectance in the GEE platform [30–32]. The date range of MODIS/Terra data was from 24 February 2000 to 31 July 2020 (https://developers.google.com/earth-engine/ datasets/catalog/MODIS_006_MOD09GQ (accessed on 31 July 2020); https://developers. google.com/earth-engine/datasets/catalog/MODIS_006_MOD09GA (accessed on 31 July 2020)). The date range of MODIS/Aqua data was from 4 July 2002 to 31 July 2020 (https: //developers.google.com/earth-engine/datasets/catalog/MODIS_006_MYD09GQ (accessed on 31 July 2020); https://developers.google.com/earth-engine/datasets/catalog/ MODIS_006_MYD09GA (accessed on 31 July 2020)).

### 2.3. Sentinel-2 Surface Reflectance

Sentinel-2 (S2) is a wide-swath, high-resolution, multispectral imaging mission with a global 5-day revisit frequency [33]. The S2 Multispectral Instrument (MSI) samples

13 spectral bands: visible and NIR (10 m spatial resolution), red edge, and SWIR (20 m spatial resolution), and atmospheric bands (60 m spatial resolution). It provides data suitable for assessing the state and change of vegetation, soil, and water cover. Observation of inland waterways and coastal areas [34–36]. The dataset has been available since 14 December 2018. The aim is to validate the water–ice classification accuracy of RF derived from MODIS, and the reconstruction results of cloud gap-filling (https://developers.google.com/earth-engine/datasets/catalog/COPERNICUS_S2 (accessed on 31 July 2020)).

### 2.4. JRC GSW

The JRC GSW dataset contains maps of the location and temporal distribution of surface water from 1984 to 2019 and provides statistics on the extent and change of those water surfaces [37]. These data were generated using 4,185,439 scenes from Landsat 5, 7, and 8 acquired between 16 March 1984 and 31 December 2019. Each pixel was individually classified into water/non-water using an expert system and the results were collated into a monthly history for the entire period and two epochs (1984–1999, 2000–2019) for change detection [38]. The aim is to generate the annual maximum boundary of Qinghai Lake and constrict the daily pixels of water and ice within the boundary (https://developers.google.com/earth-engine/datasets/catalog/JRC_GSW1_2_YearlyHistory (accessed on 31 July 2020)).

### 2.5. ERA5 Daily Aggregates

ERA5 is the fifth generation European Centre for Medium-Range Weather Forecasts atmospheric reanalysis of the global climate. Reanalysis combines model data with observations from across the world into a globally complete and consistent dataset. The two critical variables—2 m air temperature and total precipitation—are provided by ERA5 Daily Aggregates in GEE. Additionally, daily (monthly/yearly) total precipitation values are given as daily (monthly/yearly) sums; daily (monthly/yearly) temperature values are provided as daily (monthly/yearly) averages [39,40]. The objective is to analyze the relationship with the pixel wise spatial-temporal distribution of the freeze–thaw cycle of Qinghai Lake (https://developers.google.com/earth-engine/datasets/catalog/ECMWF_ERA5_DAILY (accessed on 31 July 2020)).

## 3. Methodology

### 3.1. RF Classification

The previous research compared the water index, ice threshold value method and RF algorithm. The RF is very robust and can handle complex data, and the advantage is that bagging can avoid overfitting [22]. The RF algorithm is a classification and regression algorithm originally designed for machine learning (ML) [20,41]. ML is a discipline focused on two interrelated questions: "How can one construct computer systems that automatically improve through experience?" and "What are the fundamental statistical–computational–information-theoretic laws that govern all learning systems, including computers, humans, and organizations?" [42]. Figure 2 shows the flowchart of MODIS data using RF based on the GEE platform. Individual trees are derived using a bootstrapped sample of the original data. Approximately two-thirds of the samples (referred to as "in-bag" samples) in the dataset are used for training and the remaining one-third (referred to as "out-of-bag" samples) are used in an internal cross-validation technique for estimating how well the resulting RF model performs [20]. In the GEE platform, the RF algorithm included six critical parameters, which were the number of trees, variablesPerSplit, min-LeafPoplation, bagFraction, outofBagMode, and seed variables. It excluded the criterion and maximum tree depth. The number of trees was set to 85 depending on the training data, the variable PerSplit is 0.66, other variables could be set as default, the detailed information was given in the previous research [22].

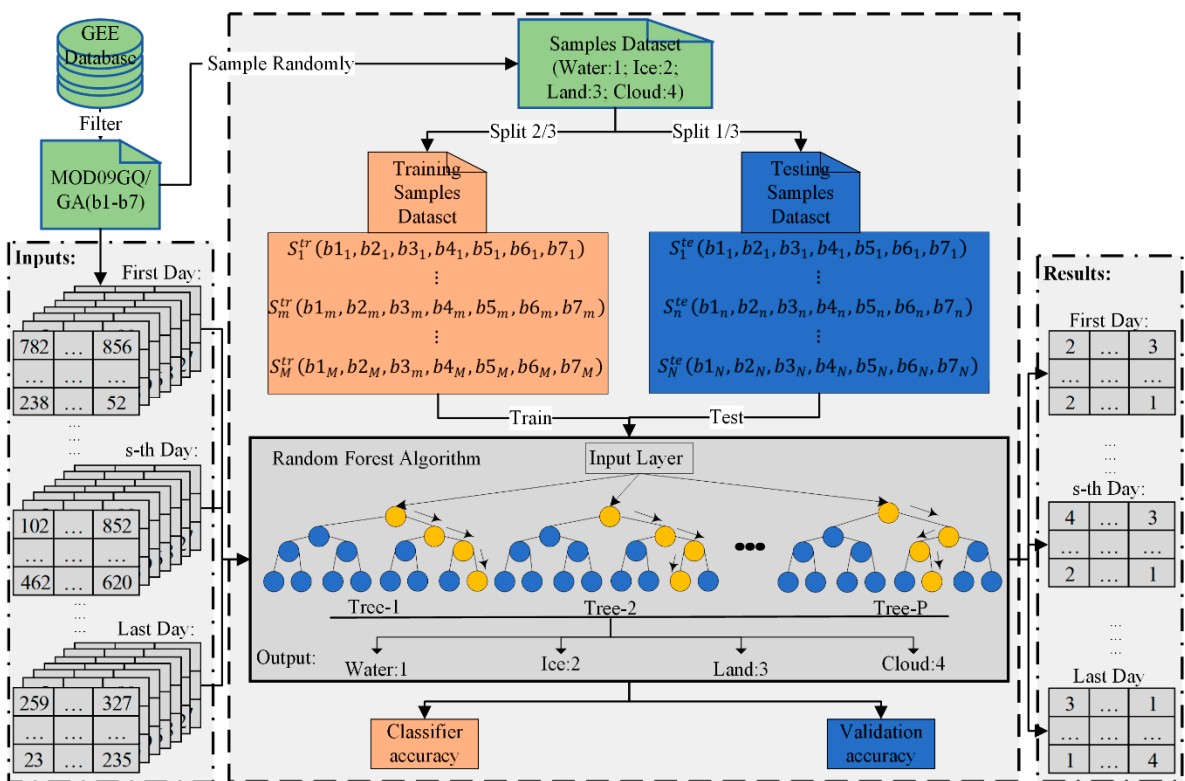

**Figure 2.** Flowchart of MODIS data using RF based on the GEE platform (left panel is the input MODIS images; right panel is the classified results; middle panel is the processes of the RF model, includes the training samples, testing samples and RF model, the number of classes is four, samples dataset includes seven bands).

### 3.1.1. Training and Testing Dataset

The training data and testing data were derived from MODIS products, which are "MOD09GQ V6" and "MOD09GA V6". As the freeze-up events of lake surface water generally occur in November, and break-up events of lake ice occur in April, the ice duration is five months [29]. Four images with MODIS Surface Reflectance in the middle of October (15 October), January (15 January), April (15 April), and July (15 July) are selected in each year, and 100 sample pixels (4 classes: water, ice, land, and cloud) are simply randomly selected in each image. As a result, 84 images were selected over 24 February 2000 to 31 December 2020. Additionally, if the MODIS images showed bad sample pixels in the seven-band features with low quality, the nearest former or following images were used instead of the current images. For example, a MODIS image showed some bad pixels on the 15 April 2001, but the pixels of the next image were fine on the 16 April 2001, which were used in place of the image on the 15 April 2001. The total number of MODIS sample pixels is 7776 after quality control, including water (2343), ice (1421), land (2850), and cloud (1162) pixels. Finally, the 2/3 (5184) sample pixels were randomly assigned to the training sample. Meanwhile, the remanent 1/3 (2592) of sample pixels were assigned to the testing sample.

### 3.1.2. Model Training and Validation

The number of RF trees is different. The results of the classifier accuracy and validation accuracy for RF are different, especially the validation accuracy. In general, the number of RF trees was empirically set to 100 [43–45]. Oil spills were classified and it was found that an ensemble of 70 trees was sufficient [46]. The number of RF trees varying from 1 to 75 based on a step of 5 was set to 50 [47].

The suitable and performable number of RF trees in GEE for this study was selected, the grid search was performed. Many trials were conducted from 10 to 500 by steps of 10 for

numberoftrees variable. The classifier accuracy was greater than 0.950. When the number of trees was greater than 100, the classifier accuracy was close to 1.000, the validation accuracy was between 0.747 and 0.807. Specifically, the focus was on the number of trees that ranged from 70 to 100, with the greatest validation accuracy value being 0.807 in 85 and 90 trees, and the classifier accuracy was 0.997. Herein, the key-dependent factors were validation accuracy and high computing performance; therefore, 85 was selected as the number of RF trees.

### 3.2. Cloud Gap-Filling

The classification of four classes was verified further; some outliers may be misclassified. There are some water and ice errors of commission. Meanwhile, there are some water and ice errors of omission. Cloud cover and cloud shadow have the greatest impact on classifying lake water and ice. Of paramount importance, the ice and water can only be found inside the lake. Therefore, to solve this problem, annual water mask products (JRC GSW) were used to filter out the outliers and obtain the lake water and ice information inside the lake from 2000 to 2020.

Suppose that in an ice season year, there are only water–ice-water transition processes at each pixel. The water pixels freeze up in fall or winter, the ice pixels break up in spring at the corresponding pixels, there are no multiple water–ice state transitions during an ice season year. The critical hypothesis conducts the annual temporal-spatial distribution of surface water freeze-up dates and ice break-up dates; some misclassifications of four classes could be reduced and eliminated.

An effective gap-filling scheme via Non-local Spatio-Temporal Filter (NSTF) is applied to reconstruct the daily classification results under the cloud cover gaps. Due to the irregular gaps under the corresponding cloud regions, the classification results derived from Terra and Aqua are blended first [48]. The adjacent temporal filtering (ATF) is executed to fill some cloud-gaps [49–54], and finally, the NSTF is applied to fill most of the remaining cloud-gap [55–57].

### 3.3. Methods Evaluation Approaches

It is critical to evaluate the water–ice classification results. The confusion matrix and the derived indexes are applied to evaluate the results (Table 1). The kappa coefficient is frequently used to summarize the accuracy assessment of land cover classifications obtained from remote sensing. The reference classification is obtained from the ground visit or other spatial high-resolution remote sensing sample data. The estimated sample date is often summarized in a confusion matrix subjected to various statistical analyses [58].

**Table 1.** Accuracy estimation and confusion matrix for all classes ($a_i$ stands for the total number of actual class i pixels, $b_j$ stands for the total number of predicted class j pixels, and $N_{ij}$ stands for the total number of actual class i and predicts class j pixels, N stands for the total number of pixels, q stands for the number of classes).

| | | Predicted | | | | Row Total |
|---|---|---|---|---|---|---|
| | | **1** | **2** | ... | **q** | |
| **Actual** | 1 | $N_{11}$ | $N_{12}$ | ... | $N_{1q}$ | $a_1$ |
| | 2 | $N_{21}$ | $N_{22}$ | ... | $N_{2q}$ | $a_2$ |
| | ... | ... | ... | $N_{ij}$ ... | ... | $a_i$ ... |
| | q | $N_{q1}$ | $N_{q2}$ | ... | $N_{qq}$ | $a_q$ |
| **Column Total** | | $b_1$ | $b_2$ | $b_j$ ... | $b_q$ | N |

Suppose N pixels of remote sensing images are classified into q categories. Given a census of N pixels and the true classification of each pixel, $a_i$ stands for the total number of actual class i pixels, $b_j$ stands for the total number of predicted class j pixels, and $N_{ij}$ stands

for the total number of actual class i and predicts class j pixels. The population confusion matrix is shown in Table 1.

The producer accuracy (PA) is the following Equation (1):

$$PA_i = \frac{N_{ii}}{a_i} \tag{1}$$

The overall accuracy (OA) is the following Equation (2):

$$OA_i = \frac{1}{N} \sum_{i=1}^{q} N_{ii} \tag{2}$$

Kappa coefficient (K) is the following Equation (3):

$$K = \frac{OA - P_e}{1 - P_e} \tag{3}$$

where the $P_e$ is the following Equation (4):

$$P_e = \frac{1}{N^2} \sum_{i=1}^{q} a_i b_i \tag{4}$$

*3.4. Definition of Lake Phenology*

For convenience, the ice season year is defined as running from 1 September to 31 August of the following year. Each pixel has its water–ice state phase of the lake. Firstly, the water froze into the solid ice state, then the ice melted into the water during the annual fall–winter–spring successive ice season for pixels. We used the general calculation method for Julian calendar dates referred to in the literature [10,29,59], i.e., when this day occurs within the year (day of the year, DOY, with 1st January as the reference), the consecutive date of the following year is added. Due to the difference of Julian calendar dates in each pixel, the spatial-temporal distribution of the freeze–thaw cycle is very critical. The Julian calendar date of the first pixel (the water froze into the solid ice state) was set as the date of freeze-up start (FUS) of the lake. That of the last pixel (the water froze into the solid ice state) was set as the date of the freeze-up end (FUE) [60,61]. Symmetrically, that of the first pixel (the ice melt into the water state) was set as the date of break-up start (BUS). The last pixel (the ice melt into the water state) was set as the date of break-up end (BUE).

Therefore, the attendant variables of lake phenology are the freeze-up duration (DFU), full ice cover duration (DFI), break-up duration (DBU), ice duration (DI), and full water duration (DFW). The DFU is the duration that ranges from FUS to FUE; the DFI is the duration that ranges from FUE to BUS; the DBU is the duration that ranges from BUS to BUE; the DI is the duration that ranges from FUS to BUE; the DFW is the duration that ranges from BUE to following FUS.

## 4. Results

*4.1. Cloud Removal from MODIS Images*

The first challenge in information extraction from optical satellite images of the Earth's surface is the collection of cloud-free images. Clouds dramatically affect signal transmission in complex ways due to their different shapes, heights, and distributions, thus contaminating remote sensing data from land and water. Cloudy image restoration is a vital step in the remote sensing image processing chain. The correction of cloudy data can substantially increase the number of useable images and pixels available for later applications. Cloud removal strives to remove cloud effects, including cloud shadows. The daily Terra/Aqua cloud was calculated using cloud state and cloud shadow bitmask of state 1km. The daily percent and daily accumulative percent with Terra/Aqua cloud/cloud shadow percent group are shown in Figure 3.

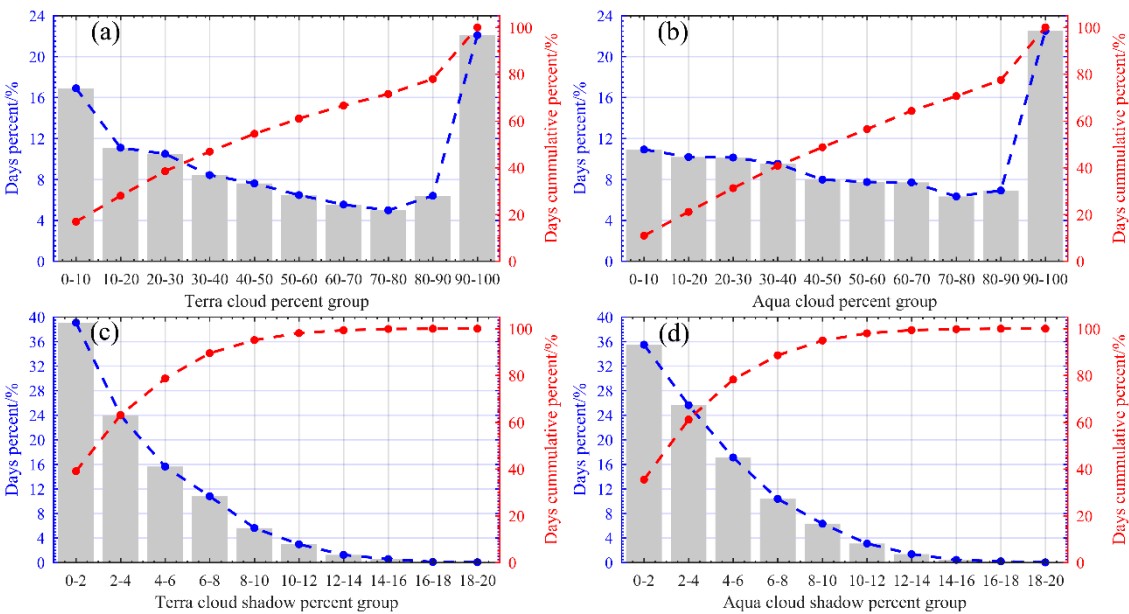

**Figure 3.** The daily percent and days' accumulative percent with the Terra/Aqua cloud/ cloud shadow percent groups ((**a**). Terra cloud percent group; (**b**). Aqua cloud percent group; (**c**). Terra cloud shadow percent group; (**d**). Aqua cloud shadow percent group).

The cloud percent covered from 0% to 100%, the range of cloud percent was divided into 10 groups by 10% step, Figure 3a,b show the days' percent and days accumulative percent with Terra/Aqua cloud percent group. The trend of days' percent with Terra/ Aqua cloud percent was similar, the daily percent decreased from the group (0, 10), reached the lowest point in the group (70, 80), and lasted the longest in the group (90, 100). However, the daily percent in Terra was greater than that in Aqua in two groups (0, 10) and (10, 20). In contrast, the daily percent in Terra was less than that in Aqua in four groups (50, 60), (60, 70), (70, 80), and (80, 90). The daily accumulative percent increased fast, then slowly in Terra; however, it increased with a relatively uniform rate in Aqua, which was about 40% when the cloud percent was less than 30% in Terra, but that reached the same level when the cloud percent was less than 40% in Aqua; that was greater than 50% in Terra, nevertheless, that was less than 50% in Aqua, when the cloud percent was less than 50%; that reached about 78% both in Terra and Aqua when the cloud percent was less than 90%.

For cloud shadow that was less influential but cannot be ignored, the cloud shadow percent covered from 0% to 20%, the range of cloud shadow percent was divided into 10 groups by 2% step. Figure 3c,d show the days' percent and days' accumulative percent with Terra/Aqua cloud shadow percent group. The trend of days' percent/days' accumulative percent with Terra/Aqua cloud percent was similar and showed exponential decrease/increase. The daily percent decreased from about 40% in the group (0, 2) in Terra, reaching the lowest point 0 in the group (18, 20). Comparably, the daily percent decreased from about 36% in the group (0, 2) in Aqua, reached the lowest point 0 in the group (18, 20). The daily accumulative percent increased fast, then slowly, both in Terra and Aqua, reached about 100% in the group (8–10). It was less influential when the cloud shadow percent was larger than 10%.

The monthly cloud and cloud shadow percent in Terra and Aqua were shown in Figure 4. The trend change of cloud percent and cloud shadow percent in Terra was similar to that in Aqua. The mean monthly cloud percent in Terra was 49.46%, less than about 4% in Aqua (53.63%). The maximum monthly cloud percent in Terra was 71.92%, which was the same as in Aqua (72.02%). The minimum monthly cloud percent in Terra was 16.63%, less than about 6% in Aqua (22.50%). It was illustrated that there were the more cloudy conditions in Afternoon (Aqua) than Morning (Terra). Simultaneously, the mean monthly

cloud shadow percent in Terra was 3.58%, less than about 0.16% in Aqua (3.74%). The maximum monthly cloud shadow percent in Terra was 7.33%, less than about 0.30% in Aqua (7.63%). The minimum monthly cloud percent in Terra was 0.70%, less than about 0.97% in Aqua (1.67%). It was the same result as the cloud, and there was more cloudy shadow in the Afternoon (Aqua) than Morning (Terra). The cloud percent was 13.82 times than cloud shadow percent in Terra, cloud shadow percent was 14.34 times than cloud shadow percent in Aqua—this was an interesting result. The cloud percent was about 14 times than cloud shadow percent. To some degree, the cloud shadow cannot make much difference to cloud removal.

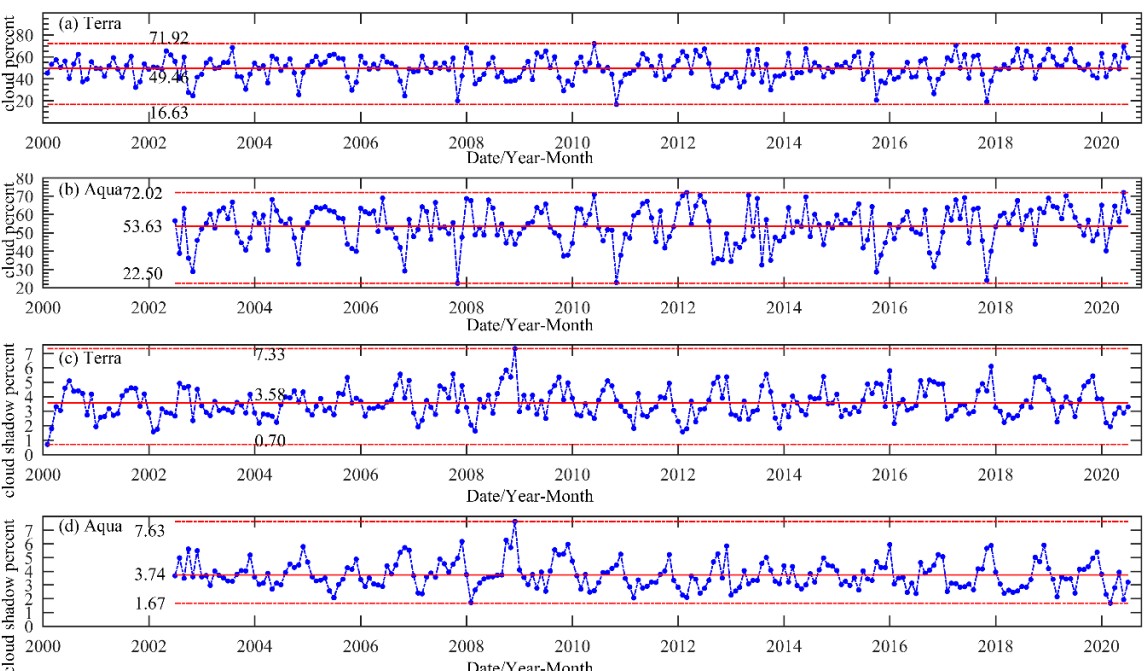

**Figure 4.** Monthly cloud and cloud shadow percent from MODIS in Terra and Aqua ((**a**). cloud percent from Terra; (**b**). cloud percent from Aqua; (**c**). cloud shadow percent from Terra; (**d**). cloud shadow percent from Aqua. The above red dotted line is the maximum, the below red dotted line is the minimum, the red solid line in the middle is the average).

*4.2. Extraction of Water and Ice by RF Algorithm*

Table 2 shows the training classification results of four classes based on the confusion matrix of RF. Four classes' results were expressed as the pixel count, the bold count stands for the predicted accurate class results, while the accuracy of RF was expressed as the percentage. The training kappa coefficient was 99.59%, the overall training accuracy of RF was 99.71%, so the agreement is almost perfect. The obtained accuracies were high for all classes. Table 3 shows the testing classification results of four classes based on the confusion matrix of RF. The overall testing accuracy was 97.45%, testing the kappa coefficient was 96.45%, the agreement is almost perfect. The accuracies were high for all four classes.

Daily water and ice classes were extracted by the RF algorithm according to trained RF classifiers, which is based on four classes of training sample pixels and testing sample pixels. Four daily scenes with water–ice-cloud mixed conditions were selected, which were from around April. Figure 5 visually listed the RF water–ice classification in Terra and Aqua under the cloud cover condition. Figure 5(1), (3) show the four daily raw scenes in Terra and Aqua. Figure 5(2), (4) show the water–ice classification using RF. The classification results were in good agreement with the raw surface reflectance image. These scenes were influenced by the cloud cover and cloud shadow, which were distributed randomly over the lake. Scenes were slightly influenced by the cloud cover and cloud shadow on 6 April 2018 (Figure 5d) and the water–ice extraction was slightly influenced. Scenes were

slightly influenced by the cloud cover and cloud shadow outside the lake on 24 March 2014 (Figure 5c) but water–ice extraction was relatively complete. The more cloudy conditions are distributed near the lake on 11 April 2004 (Figure 5a), and the water–ice classifications were partly influenced. Dense cloud covered almost the whole lake on 15 April 2008 (Figure 5b), and the water–ice classifications were severely affected. The images from Aqua were more likely affected than the images from Terra; these results were in good agreement with the previous daily distribution of cloud and cloud shadow. Although the images from Aqua and Terra were affected by the cloud and cloud shadow, the result could be completely reconstructed by gap-filling procedures after the RF classification of images.

**Table 2.** Training confusion matrix and accuracy of RF for four classes. (Bold font is the number of pixels of predicted values consistent with the actual values).

| | | Predicted | | | | Producer Accuracy/% |
|---|---|---|---|---|---|---|
| | | **Water** | **Ice** | **Land** | **Cloud** | |
| **Actual** | Water | **1538** | 7 | 0 | 0 | 99.55 |
| | Ice | 1 | **909** | 0 | 1 | 99.78 |
| | Land | 1 | 0 | **1872** | 2 | 99.84 |
| | Cloud | 0 | 2 | 1 | **780** | 99.62 |
| | | Overall accuracy: 99.71 | | | | |
| | | Kappa coefficient: 99.59 | | | | |

**Table 3.** Testing the confusion matrix and accuracy of RF for four classes. (Bold font is the number of pixels of predicted values consistent with the actual values).

| | | Predicted | | | | Producer Accuracy/% |
|---|---|---|---|---|---|---|
| | | **Water** | **Ice** | **Land** | **Cloud** | |
| **Actual** | Water | **776** | 18 | 3 | 1 | 97.24 |
| | Ice | 3 | **491** | 0 | 16 | 96.27 |
| | Land | 4 | 4 | **960** | 7 | 98.46 |
| | Cloud | 1 | 9 | 2 | **367** | 96.83 |
| | | Overall accuracy: 97.45 | | | | |
| | | Kappa coefficient: 96.45 | | | | |

To validate the water–ice classification accuracy of RF derived from MODIS, 2979 validation samples from Sentinel-2 images were selected manually from 14 December 2018 to 29 December 2019. We selected 150 composite images and an average of 20 sample points were sampled for each composite image. We obtained 2377 matched samples through the validation procedure, including the time matching and space matching for RF water–ice results and validation samples. Table 4 shows the validation accuracies and confusion matrix. The obtained validation accuracies were 98.50% and 97.62% for the water and ice classes, respectively. The overall validation accuracy was 98.36%, the validation kappa coefficient was 94.00%—the agreement is almost perfect.

**Table 4.** Confusion matrix and validation accuracy for water–ice classes. (Bold font is the number of pixels of predicted values consistent with the actual values).

| | | Predicted | | Producer Accuracy/% |
|---|---|---|---|---|
| | | **Water** | **Ice** | |
| **Actual** | Water | **1969** | 30 | 98.50 |
| | Ice | 9 | **369** | 97.62 |
| | | Overall accuracy: 98.36 | | |
| | | Kappa coefficient: 94.00 | | |

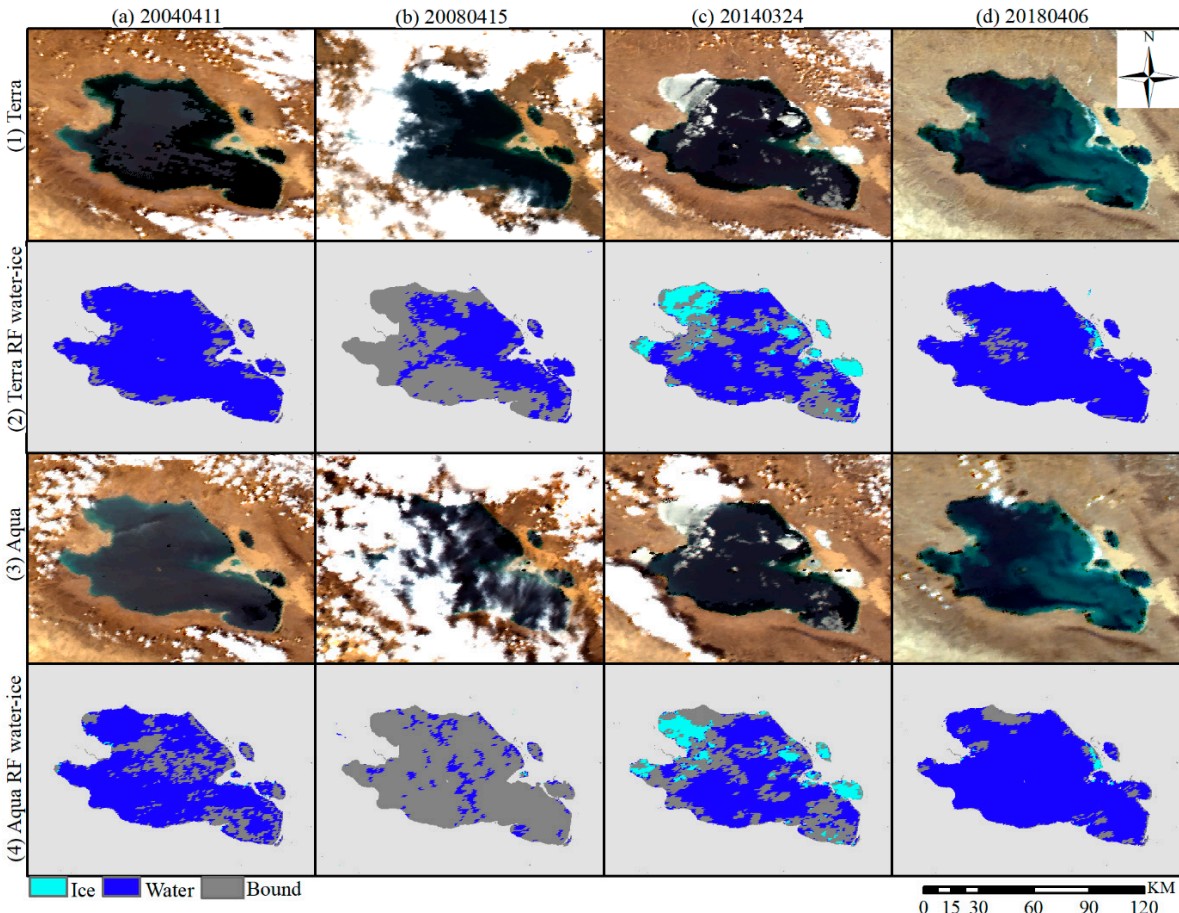

**Figure 5.** Cloud removal and RF water–ice classification in Terra and Aqua. (Note: This figure is created using MODIS and Google Earth Engine, (**a**–**d**) is the date).

### 4.3. Reconstruction of Water and Ice by Cloud Gap-Filling

An effective gap-filling scheme was applied via a combination of Terra and Aqua, ATF, and NSTF methods to reconstruct the daily classification results of Terra and Aqua under the cloud cover gaps. The results of three cloud gap-filling procedures are visually illustrated in Figure 6. Figure 6 mainly illustrates the two critical liquid water-to-solid-ice state or solid-ice-to-liquid-water state transfer processes in the co-existence of Terra and Aqua: liquid water freeze-up process (Figure 6) and solid ice break-up process (Figure 7).

Figure 6 (1), (2) show the RF ice-water classification results of Terra and Aqua during the 25 December 2002 to 30 December 2002. The ice was freezing up in these data ranges, but the cloud gaps seriously influenced the extraction of ice and water, so complete ice and water pixels were severely missing. Figure 6 (3) combined the Terra and Aqua to compensate for the shortcomings of the single satellite. The ATF method procedure (Figure 6 (4)) could restore the most missing cloud gap data. The NSTF method procedure could completely recover the data. Furthermore, Figure 7 shows the ice break-up process on 27 March 2003 to 1 April 2003. Figure 7e shows the full cloud cover, but the cloud gap was completely reconstruct through the cloud gap-filling procedures. Figure 7a,f were severely covered by cloud. The cloud gap could be restored. The visualized results showed the feasibility and superiority of cloud gap-filling procedures.

To validate the performance of the cloud gap-filling procedures, the validation samples from Sentinel-2 images were used to quantitatively analyze the overall accuracy of the cloud gap-filling procedures. Table 5 shows the validation accuracies and confusion matrix. The obtained validation produced accuracies of 90.65% and 97.94% for the water and ice

classes, respectively. The overall validation accuracy was 92.56%, the validation kappa coefficient was 82.17%, the agreement is almost perfect.

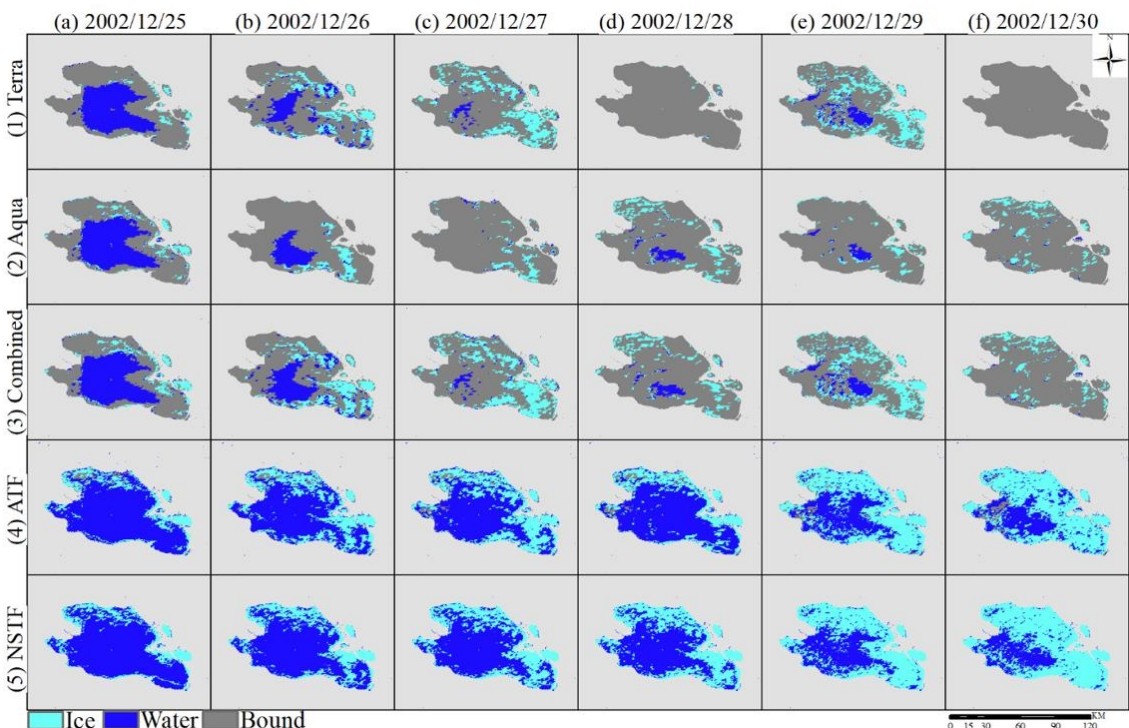

**Figure 6.** Cloud gap-filling procedure of ice freeze-up process from 25 December 2002 to 30 December 2002. (Note: This figure was created using MODIS and Google Earth Engine, (**a**–**f**) are 25 December 2002, 26 December 2002, 27 December 2002, 28 December 2002, 29 December 2002, 30 December 2002, respectively).

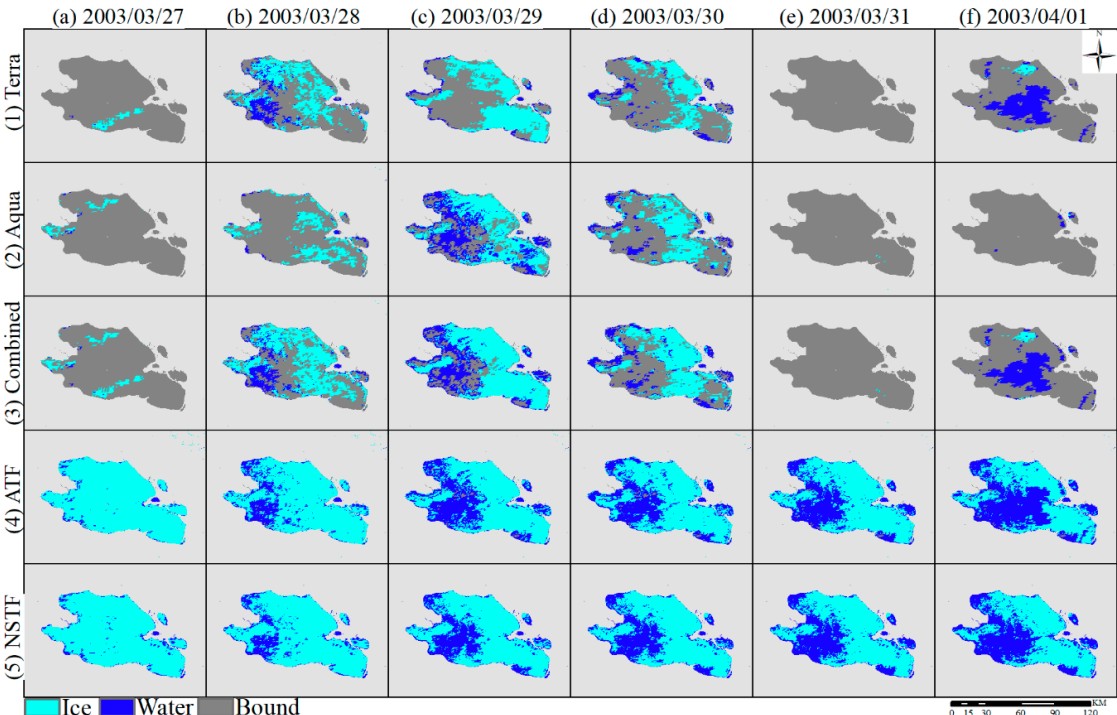

**Figure 7.** Cloud gap-filling procedure of ice break-up process during 27 March 2003 to 1 April 2003. (Note: This figure was created using MODIS and Google Earth Engine, (**a**–**f**) are 27 March 2003, 28 March 2003, 29 March 2003, 30 March 2003, 31 March 2003, 1 April 2003, respectively).

**Table 5.** Validation accuracy and confusion matrix for the gap-filling procedure. (Bold font is the number of pixels of predicted values consistent with the actual values).

|  |  | Predicted | | Producer Accuracy/% |
|---|---|---|---|---|
|  |  | **Water** | **Ice** |  |
| **Actual** | Water | **1978** | 204 | 90.65 |
|  | Ice | 16 | **760** | 97.94 |
| | | Overall accuracy: 92.56 | | |
| | | Kappa coefficient: 82.17 | | |

### 4.4. Annual Spatial Dynamics of Freeze-Up and Break-Up Dates

The previous work mainly studied the lake ice as a whole, while this work focused on each pixel of the lake for detailed research. The lake images are composed of water–ice pixels. Each pixel has its own water–ice state phase, with either solid ice or liquid water, and each pixel transitions between the above two states: firstly, the liquid water was transited into the solid ice state. The ice melted into the water during the annual fall–winter–spring successive ice season. Therefore, the date on which each pixel transitions from water-to-ice during the surface water freeze-up processes or from ice-to-water during the ice break-up processes is crucial in each ice season year. Figure 8a visualized the annual spatial distribution of the ice freeze-up dates in each pixel on the Julian calendar day during the 2000/2001–2019/2020 ice season years. Generally, the freeze-up date ranged primarily from Julian calendar day 330 to day 397. The earliest positions were located on the northern and western parts along the lake shoreline. The surface water was frozen from along the lake boundary to the middle, and the last freezed positions were located on the southern and central parts of the lake. The lake water freeze-up rule was also very consistent with the latitude range of Qinghai Lake.

Figure 8b,c illustrate the temporal variation and temperature of the annual ice freeze-up at each pixel. The earliest cold winter came in the 2012/2013 ice season, with the greatest number of positions of approximately 335 (1 December), and a few were approximately 370 (5 January). The next earlier cold winter was during the 2005/2006 ice season. The latest cold winter came in the 2016/2017 ice season; the overwhelming majority cover positions were larger than 375 (10 January), there were approximately 380 (15 January) in the south and along the southern boundary, and very few were approximately 370 (5 January), which started the freeze-up. The latest cold ice season was 2015/2016. The fastest ice freeze-up year was the 2019/2020 ice season. It lasted about 5 days. The majority of pixels ranged from DOY 365 to 369. The slowest ice freeze-up year was the 2000/2001 ice season. It lasted about 15 days and the majority of pixels ranged from DOY 352 to 366.

The ice melted into the water when the spring was coming. Figure 9a depicts the annual spatial distribution of the ice break-up date for each pixel on Julian calendar days during the 2000/2001–2019/2020 ice season. The break-up dates ranged primarily from Julian calendar day DOY 70 to 116, and lasted average 46 days. As a whole, the ice began to break up from the southwestern parts or center of the lake gradually diffused to the northern and eastern lake boundary, the coastal area and several bays ultimately melted.

Figure 9b,c depict the temporal variation and temperature of the annual ice break-up at each pixel. The earliest warm spring year was in 2013/2014, where the 1/3 ice-to-water pixels ranged from DOY 75 to 80, and nearly all pixels began to melt before DOY 83 (24 March). A warmer spring year occurred in 2015/2016, the overwhelming majority of ice-to-water pixels ranged from DOY 80 to 84. The latest cold spring was the 2007/2008 ice season year, a far-reaching 2008 spring snow disaster took place in China, and the ice break-up of all lake pixels was delayed—the Julian calendar day ranged from DOY 101 to 105. The coldest spring was the 2010/2011 ice season year, the ice break-up of all lake pixels ranged from DOY 97 to 107. The fastest ice break-up year was the 2012/2013 ice season; it lasted about 4 days and the majority of pixels ranged from DOY 92 to 95. The

slowest ice freeze-up year was the 2011/2012 ice season; it lasted about 33 days and the majority of pixels ranged from DOY 71 to 103.

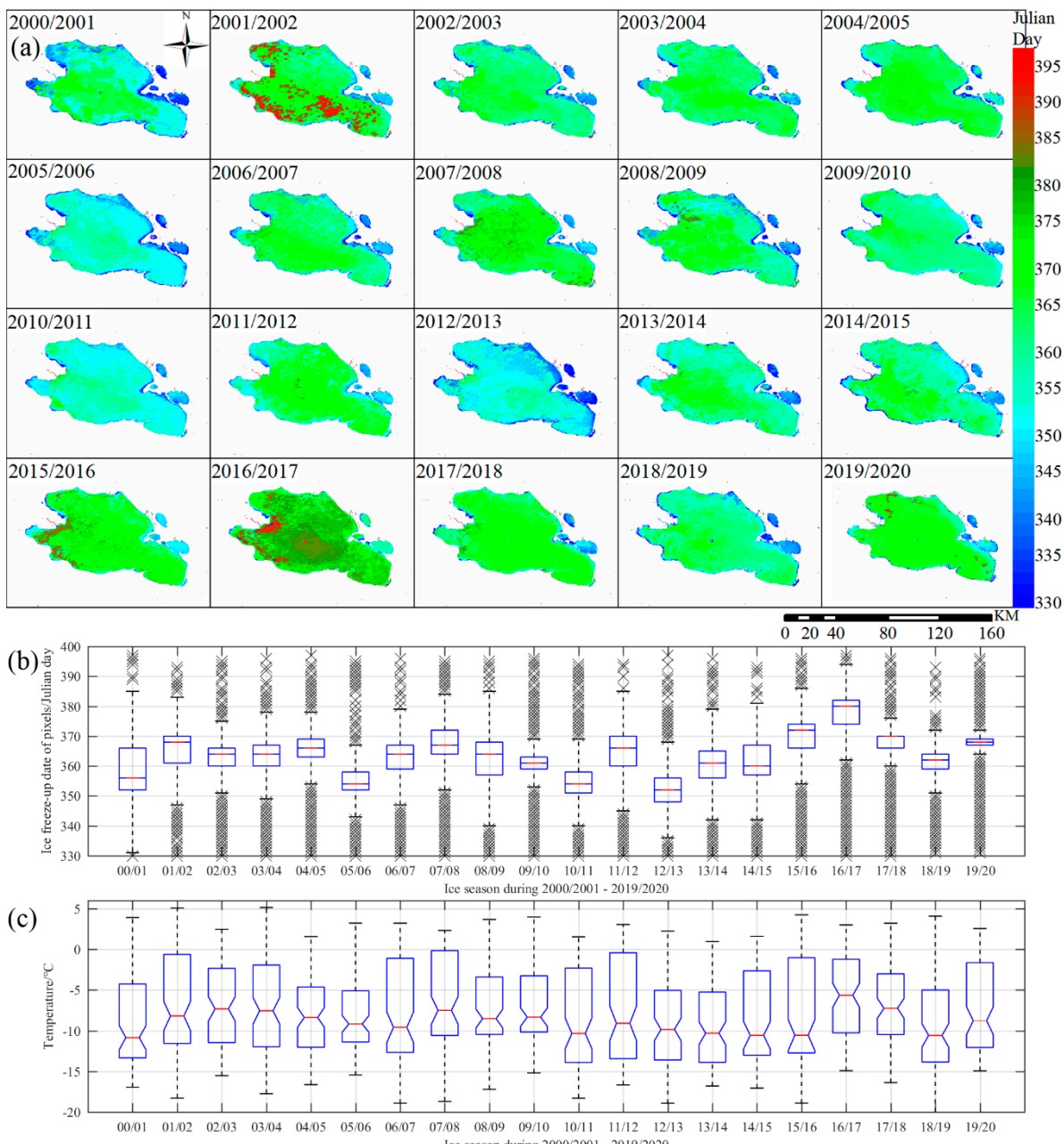

**Figure 8.** The Julian calendar day of ice freeze-up pixel-by-pixel during the 2000/2001–2019/2020 ice season (00/01 indicates the 2000/2001 ice season year, other symbols have the same meanings—(**a**). the ice freeze-up date at each pixel; (**b**). the boxplot of ice freeze-up by Julian calendar day; (**c**). the boxplot of the temperature). (Note: This subfigure was created using MODIS and Google Earth Engine).

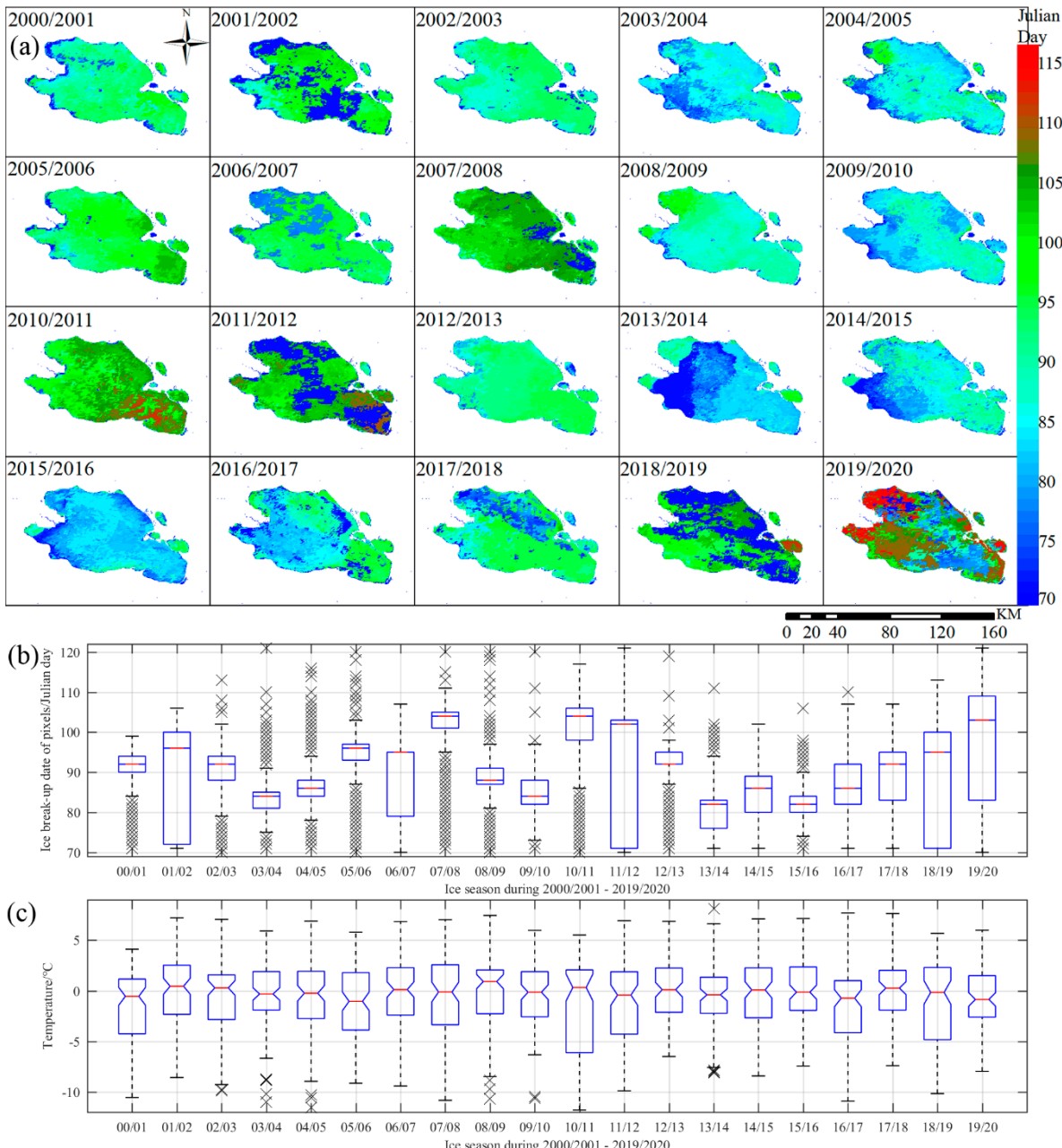

**Figure 9.** The Julian calendar day of ice break-up pixel-by-pixel during the 2000/2001–2019/2020 ice season ((**a**). the ice break-up date at each pixel; (**b**). the boxplot of ice break-up Julian calendar day; (**c**). the boxplot of temperature). (Note: This sub-figure (**a**) is created using MODIS and Google Earth Engine).

Distinctively, the 2019/2020 ice season year also saw the coldest spring; the ice break-up of all lake pixels ranged from DOY 83 to 109, which lasted a longer time than other ice season years. This result may have a close correlation with precipitation and temperature. Figure 10 shows the daily air temperature at 2 m and precipitation derived from the ERA5 during the solid ice break-up process in the 2019/2020 ice season. It snowed four times during this period: 15 March; 25 March to 7 April; 16 April to 17 April; and 20 April to 24 April at the same time, the snowfall was accompanied by four distinctly advanced temperature drops, respectively. The difference was that the temperature was greater than 0°C after the third snowfall events (14 April). Therefore, snowfall and adjoint cooling would result in the continuous existence of lake ice, which would not melt at the earlier

time as in previous years, presenting a unique spatial distribution phenomenon. Therefore, the lake phenology is closely related to precipitation and temperature.

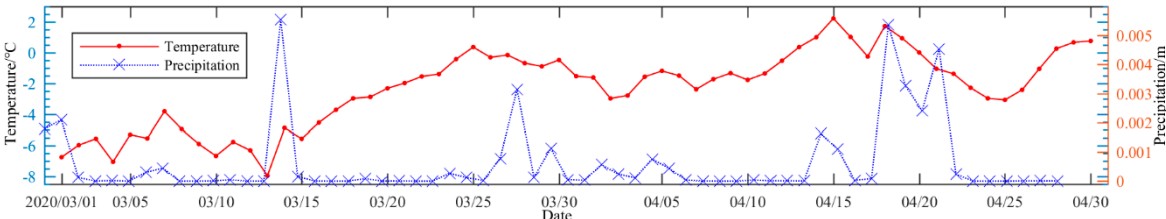

**Figure 10.** The daily precipitation and 2 m air temperature during the ice break-up process in the 2019/2020 ice season.

## 5. Discussion

The variation of the liquid water-to-solid-ice state and the solid-ice-to-liquid-water state of the lake is continuous in temporal range. Figure 11 illustrates the temporal variation of the freeze–thaw-cycle processes, precipitation, and the air temperature at 2 m of Qinghai Lake, during the 2000/2001–2019/2020 ice season years. Consequently, the duration and variation trends of the DFU, DFI, DBU, and DI are very important for freeze–thaw cycle lakes; the relationships between those durations, precipitation, and temperature are very critical. Figure 11a shows the annual certain date ranges of liquid surface water freeze-up, full solid ice cover, solid ice break-up and ice period. The date of surface water freeze-up mainly ranged from 1 December to 10 January. The ice break-up mainly ranged from 10 March to 20 April, the full ice cover was centrally distributed within the embedded FUE and BUS, as well as which the ice period extended the whole lifetime of ice cover. The FUS and FUE showed a tendency to delay, while the BUS and BUE tended to advance. Most of the former studies of lake ice had also demonstrated a global warming tendency, and the latter was more intense change than the former [59,62–65].

Figure 11b illustrates the duration and change trend of four critical ice duration processes. The mean durations of DFU, DFI, DBU, and DI were 29.05, 65.95, 26.30, and 121.30 days, respectively. The annual durations fluctuated around the average, and the maximum amplitude durations were 18.95, 30.95, 24.70, and 19.3 days, respectively. Furthermore, three variations—DFU, DFI, and DI—presented a decreasing tendency, the decrease rates were −0.37, −0.34, and −0.13 days/year, respectively, and the durations were shortened by 7.4, 6.8, and 2.6 days in the past 20 years. Only the DBU presented an increasing tendency; the increasing rate was 0.58 days/yr. and the duration was lengthened by 11.6 days in the past 20 years. It is consistent with the results that the ice duration shortened 17.5 days/decade for Canada's lakes during 1985–2004 [3].

Figure 11c,d illustrate the yearly and monthly precipitation and temperature during the 2000/2001 to 2019/2020 ice season years. The mean monthly cumulative precipitation is 0.056 m, the monthly increase rate is $4.06 \times 10^{-5}$ m/mo., the maximum monthly precipitation is gradually increasing. The maximum yearly precipitation is gradually increasing, the mean yearly cumulative precipitation is 0.681 m, the yearly increase rate is $7.700 \times 10^{-3}$ m/year, and the precipitation has increased by 154 mm in the past 20 years. It is very similar to the above temperature result. The mean monthly temperature is 1.568 °C, the monthly increase rate is $2.200 \times 10^{-3}$ °C/mo.; thus, the monthly temperature is gradually increasing. Meanwhile, the mean yearly temperature is 1.691 °C, and the yearly increase rate is 0.118 °C/yr., meaning that the temperature has increased by 2.360 °C in the past 20 years. The temperature began to increase during the 2004/2005–2010/2011 ice season years and the precipitation started to increase during the 2002/2003–2008/2009 ice season years. The relationship between the Polish lowland lakes' ice and winter air temperature was analyzed, the ice cover duration has been becoming shorter under the climatic changes condition [62].

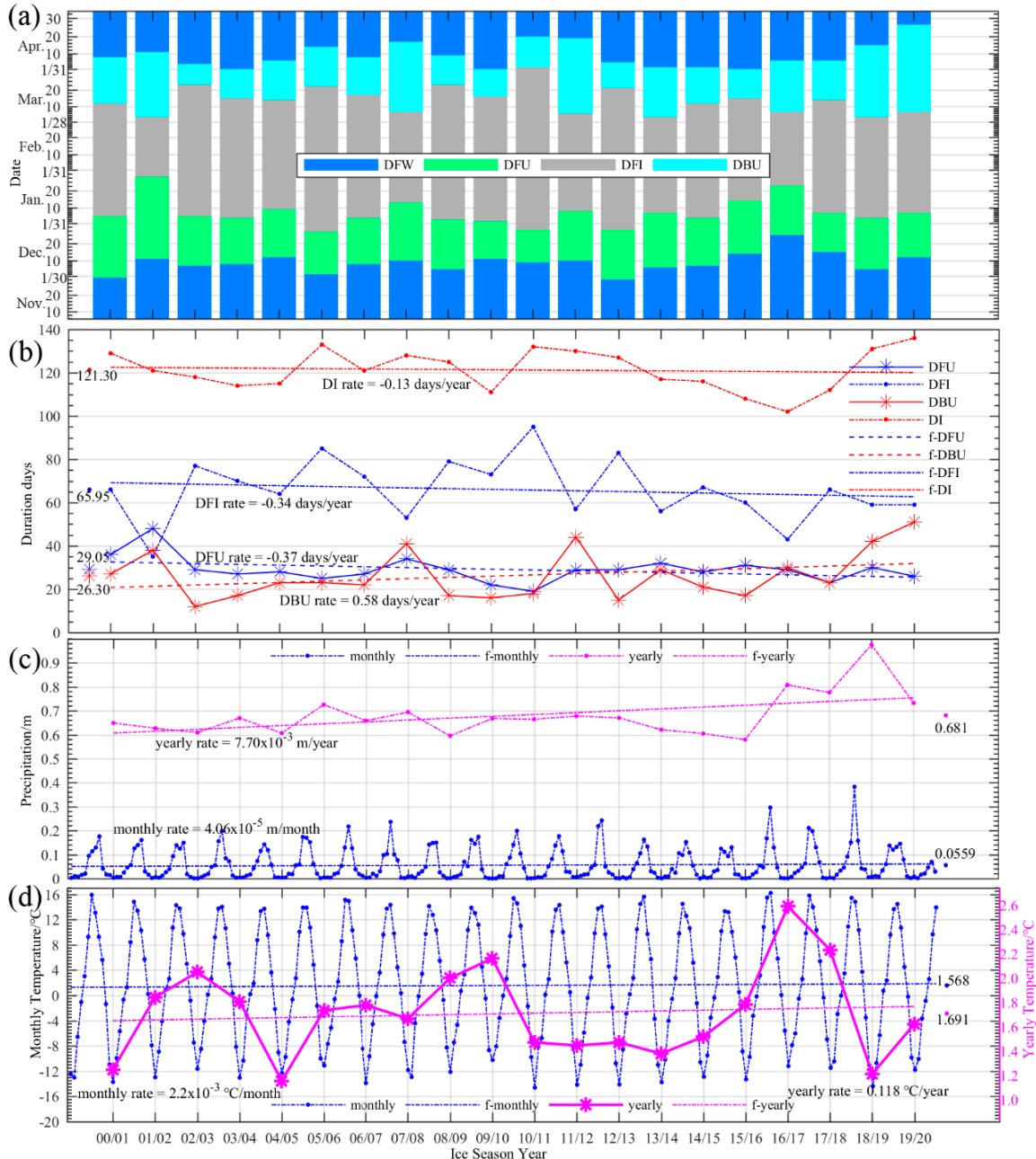

**Figure 11.** The comparison between days in duration and variation trend of annual DFU, DFI, DBU, DI, and DFW, precipitation and 2 m air temperature during 2000/2001–2019/2020 ice season ((**a**). the freeze–thaw cycle, the y axis scale is 1 (dense) or 2 (sparse) days; (**b**). days in duration and variation trend, f-DFU represented the fitted DFU, (**c**). precipitation; (**d**). 2 m air temperature, f-DFU stands for the fitted line of DFU; f-DBU stands for the fitted line of DBU; f-DFI stands for the fitted line of DFI; f-DI stands for the fitted line of DI; f-monthly stands for the fitted line of monthly value; f-yearly stands for the fitted line of yearly value).

Furthermore, the main factors affecting the duration are precipitation and temperature [66]. The relationships between the duration and variation trend of annual DFU, DFI, DBU, DI, precipitation, and 2 m air temperature during the 2000/2001–2019/2020 ice season years are critical, and it was found that the trends of these variables were in relative agreement. Compared with the temperature, the precipitation showed a delayed increasing pattern of about two years. The changing trend of yearly precipitation was similar to that of DBU and DFU. Compared with the precipitation, the durations of DI showed a delayed trend pattern of about one year, although DFU and DI were negative. Compared with the

duration of DFI, the DFU showed an opposite trend pattern. The duration of DI showed a similar trend pattern. In addition, the lake size, snowfall (area and thickness) and ice thickness displayed the significant role in regulating the ice phenology [11]. Humidity is a very sensitive variable that responds to changes in lake phenology and its surrounding environment [67,68]. The theoretical consistency in geophysical data analysis from the fractals to the stochastics method was also critical [69–72].

In summary, the increase in the temperature resulted in an increase in precipitation after two years. The increase in precipitation resulted in the increase in the duration of DBU and the decrease in DFU in corresponding years, and a decrease in the duration of DI and DFI after one year. In addition, the border shape, salinity, depth and wind speed are also the impacting factors of lake ice phenology.

## 6. Conclusions

This research focuses on the spatial-temporal distribution of the freeze–thaw cycle pixel-by-pixel in Qinghai Lake based on the ML and MODIS during the 2000/2001 to 2019/2020 ice seasons. After the daily cloud and cloud shadow were extracted and removed, the water–ice pixel state results were classified using the RF algorithm and were validated using Sentinel-2 images. The combination of Terra and Aqua, ATF, and NSTF cloud gap-filling procedures were applied to fill the cloud gap, and the Sentinel-2 images were used to validate the results of the cloud gap-filling. The results showed that the accuracy of the water–ice classification by RF was 98.36%, the kappa coefficient was 94.00%. The overall accuracy of the reconstruction of water and ice by cloud gap-filling procedures was 92.56%, the kappa coefficient was 82.17%. The annual spatial dynamics of liquid water freeze-up date showed that it ranged primarily from DOY 330 (25 November) to 397 (1 February) and the solid ice break-up date ranged primarily from DOY 70 (11 March) to 116 (26 April). Meanwhile, the decrease rate of DFU, DFI, and DI were 0.37, 0.34, and 0.13 days/yr. and the duration days were shortened by 7.4, 6.8, and 2.6 days in the past 20 years. Only the rate of DBU increased, by 0.58 days/yr., and the duration days were lengthened by 11.6 days in the past 20 years. Compared with the temperature, an increase in the temperature resulted in an increase in precipitation after two years, the increase in precipitation resulted in the increase in duration days of DBU and decrease in duration days of DFU in current years, and a decrease in duration days of DI and DFI after one year.

In conclusion, the performance of the image classification of ML is efficient, and the cloud gap-filling procedures could reconstruct the spatio-temporal dynamics of the water–ice state of the lake under the complex cloud cover conditions. However, there are some limitations, such as that the spatial resolution is 250 m, and the size of the lake is relatively large. Further research will focus on the fusion of MODIS (250 m), Landsat (30 m), Sentinel-2 (10 m), and Synthetic Aperture Radar remote sensing data; it has great potential to improve spatial resolution to 10 m.

**Author Contributions:** Conceptualization, W.H. and C.H.; methodology, W.H.; software, W.H.; validation, W.H.; formal analysis, W.H. and J.H.; investigation, W.H.; resources, C.H.; data curation, W.H.; writing—original draft preparation, W.H.; writing—review and editing, C.H., J.G., J.H. and Y.Z.; visualization, W.H.; supervision, C.H.; project administration, C.H. and J.H.; funding acquisition, C.H., J.G. and J.H. All authors have read and agreed to the published version of the manuscript.

**Funding:** This work is supported by the Strategic Priority Research Program of the Chinese Academy of Sciences "CAS Earth Big Data Science Project" (Project No. XDA19040504) and the National Natural Science Foundation of China under Grants (Project No. 41801271).

**Acknowledgments:** We are grateful for the use of Google Earth Engine, Sentinel-2, JRC GSW, ERA5 and MODIS data in this study.

**Conflicts of Interest:** The authors declare no conflict of interest.

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
