# Peer review of "Spatial-Temporal Distribution of the Freeze–Thaw Cycle of the Largest Lake (Qinghai Lake) in China Based on Machine Learning and MODIS from 2000 to 2020"

_remotesensing, doi:10.3390/rs13091695_

Round 1

Reviewer 1 Report

Please see the attached file for comments.

Author Response

Dear Editors and Reviewers,

Thank you for your letter and comments concerning our manuscript entitled "Spatial-temporal distribution of the freeze-thaw cycle of the largest lake (Qinghai Lake) of China based on the machine learning and MODIS from 2000 to 2020 " (remotesensing-1111572). The comments were valuable for improving our manuscript and guiding our research. We have studied the comments carefully and have made revisions accordingly. We hope that the revisions meet with your approval. The revised portions are marked in red font in the manuscript. Detailed point-by-point descriptions of the revisions are provided below. [email protected]

Point 1: This paper focuses on the ice phenology using machine learning. More descriptions about the literature reviews for ice phenology using machine learning should be added in Section“Introduction”. In addition, which parts are novel in this paper? More detailed descriptions about the contributions of this paper would be added in the manuscript.

Response 1: Thanks for the comments, the more descriptions about the literature reviews for ice phenology using machine learning has been added in Section“Introduction”. The lines 50-54 showed “The lake ice phenology of Nam Co (Central Tibetan Plateau) derived from multiple MODIS data products was interpreted [10]. The role of climate and lake size in regulating the ice phenology of boreal lakes was analyzed [11]. The lake ice phenology was extracted using the convolutional neural network method [12]”. The novel part is in cloud gap-filling and reconstruction of water and ice by cloud gap-filling. The corrections are shown on lines 50-54.

Point 2: Authors employed four evaluation criteria (i.e., Equations (1) ~ (4)) to assess the accuracy of proposed method. As shown in Lines 218 ~ 230, what does these variables (i.e., Nii, ai, bi) stand for?

Response 2: Thanks for the comments. We have modified according to specific comments. The sentence “ai stands for the total number of actual class i pixels, bj stands for the total number of predicted class j pixels, and Nij stands for the total number of actual class i and predicted class j pixels.” was added on lines 233-235. The corrections are shown on lines 233-235.

Point 3: In Section 4.5, authors presented the temporal dynamics analysis and proposed the conclusion, indicating “the increase in the temperature, resulting in an increase of precipitation after two years”. How to link this conclusion with Figure 11?

Response 3: Thanks for the comments, we have modified according to specific comments. The sentence “The temperature began to increase in 2004/2005, 2010/2011 ice season year, the precipitation started to increase in 2002/2003, 2008/2009 ice season year” was added on lines 519-520 to explain the conclusion “the increase in the temperature, resulting in an increase of precipitation after two years”.

Point 4: The RF algorithm was used for classification problem in this paper. Could authors describe how to perform hyper-parameters such as grid search, random search or other algorithms?

Response 4: Thanks for the comments. We have modified according to specific comments. The sentence “The critical parameters of RF algorithm are the number of trees, variablesPerSplit, min-LeafPoplation, bagFraction, outofBagMode, and seed variables. The number of trees de-pends on the training data, the variablesPerSplit is 0.66, other variables could set default.” was added on lines 159-162.

The sentence “The grid search was performed, the previous article has set parameter in detail [44].” was added on lines 189-190.

Point 5: There are other machine learning techniques such as SVM or ANN. However, authors used RF. What motivates this choice?

Response 5: Thanks for the comments, We have added the sentence “Based on the comparative analysis of water index, ice threshold value method and RF algorithm in the previous research [19], the latter is very robust and can handle complex data, the main advantage is the bagging to avoid overfitting.” on lines 151-153 to explain the motivation using RF, RF method is a continuation and expansion of previous research work.

Reviewer 2 Report

Dear authors and Editor,

there is a similar article by the same authors in Remote Sensing (DOI: 10.3390/rs12244098). The current submission is titled ‘Spatial-temporal distribution of the freeze-thaw cycle of the largest lake (Qinghai Lake) of China based on the machine learning and MODIS from 2000 to 2020’ (I will call it A), while the previous paper it titled ‘Lake phenology of freeze-thaw cycles using random forest: a case study of Qinghai Lake’ (I will call it B). I don’t see any differences between the two titles, but anyway, I read the previous paper to see what might be different in this new one. Note that this new submission does not cite the previous one; but why did you publish a paper that you are not citing in your next similar paper? Usually we publish an article and then we cite this previous article to show where we paused our research, which is now advancing through the new article.

Anyway, the study aims are the following;

A: This work focuses on the ice phenology of Qinghai Lake based on MODIS images, Joint Research Center’s Global Surface Water (JRC GSW), and ERA5 dataset using the RF algorithm in the GEE platform.

B: The objective of the study was establishing or characterizing the lake phenology of Qinghai Lake based on MODIS remote sensing data using the RF algorithm in the GEE platform. What am I missing?

From the materials and methods I understand the following;

A: MODIS satellite images of the Qinghai Lake from 2000 to 2020 were acquired. An RF algorithm was trained and validated to analyze these images and discriminate ice and water areas across the lake from 2000 to 2020. The validated model was then used to actually predict the freeze-thaw cycles in the lake from 2000 to 2020 after removing clouds and filling the cloud-induced gaps.

B. MODIS satellite images of the Qinghai Lake from 2000 to 2018 were acquired. An RF algorithm was trained and validated to analyze these images and discriminate ice and water areas across the lake from 2000 to 2018. The validated model was then used to actually predict the freeze-thaw cycles in the lake from 2000 to 2018 after removing clouds and filling the cloud-induced gaps, and the results are compared with other algorithms/models.

A different approach is applied between A and B to validate the RF algorithm. A: Sentinel-2 reflectance + JRC GSW + ERA5 data + cloud removal/gap-filling; B: MODIS reflectance imagery + cloud removal/gap-filling.

So, after reading both papers, I can’t see why a new paper is necessary to describe the freeze-thaw cycles in the Qinghai Lake for pretty much the same period. I suggest that the authors should decide on the focus of the paper. Will the focus be to compare between different RF optimization schemes, different training and validation options using the Qinghai Lake as an example? Then re-write this paper as a follow-up comparison with the previous paper, apply much detail on what this new optimized approach included and what parameters were additionally used or adjusted to increase testing accuracy. Will the focus be on describing the spatiotemporal variation of the freeze-thaw cycles in the lake? Then re-write the paper without all the technical detail and focus on clearly describing the freeze-thaw cycles. But wasn’t this the aim of the previous paper? Anyway, from Figures 8 and 9, it is not clear when freezing and thawing occurs. The new information that I can see is section 4.5, which however is actually existing information (it is already there in the previous paper) presented in a different style (instead of freeze-up start and end, presented as ice duration etc.).

Other comments

Images and tables should be self-explained (i.e. read and understood without help from the main text).

Throughout the text, replace ‘Figure XX showed’ by ‘Figure XX shows’. The same goes for the tables.

Figure 11. The abbreviations DFU, DFI, DBU, DI and DFW should be explained in the caption. What is f-yearly, f-monthly, f-DBU etc.? in Figure 11a, what kind of y axis scale is this? 10-20-Dec-10-20-Jan etc. Add another column to the left and write the month, and change this column to 10-20-30-10-20-30 … In Figure 11b you say ‘duration days/days’, is this correct? Do you mean ‘duration days (days)’? Then there is no need for ‘(days)’. Precipitation/m should be obviously written as ‘Precipitation (m)’. Does the pink color represent yearly precipitation or monthly? Similarly, the blue color is probably daily precipitation? Since you have values between months it can’t be monthly.

Results and Discussion. But actually, there is no discussion. You should show the results and then discuss them based on previous literature to try to find the factors that produced these trends. You can compare your results for this lake with possible results from other lakes in China to identify common trends etc. If the article’s focus is the algorithms and their optimization, you could compare with other studies around the world that apply other algorithms and discuss on the predictive accuracy etc. But now you only show us the results. It would be better to divide Results and Discussion to two different sections.

Line 394. Are you sure about the Julian Days? If you convert 1/1/2001 to the Julian Day here https://www.aavso.org/jd-calculator it does not produce the values you mention. See here also https://sciencing.com/calculate-julian-date-6465290.html.

Line 410. What is the difference between B: Figure 7 and Table A3, and A: Figures 8 and 9? I mean, they show the freeze-up and break-up dates but are these dates different between A and B? Did the optimized RF algorithm applied in B produce different freeze-thaw periods than the RF algorithm in A? From the figures this is not clearly visible.

To conclude, I think that the focus of this paper should be clearly re-stated, whether it would be the optimization process itself or the description of the spatiotemporal variation of freeze-thawing processes in the lake (but the freeze-thawing processes have been described in the previous paper, haven’t they?). The manuscript should thus be re-written based on the selected focus as I also stated above.

Ultimately, I think that section 4.5 alone, in combination with a new RF optimization, cannot justify a new publication. This is however my personal, subjective opinion; maybe the other reviewers do not see an issue here.

Kind regards,

Author Response

Dear Editors and Reviewers,

Thank you for your letter and comments concerning our manuscript entitled "Spatial-temporal distribution of the freeze-thaw cycle of the largest lake (Qinghai Lake) of China based on the machine learning and MODIS from 2000 to 2020 " (remotesensing-1111572). The comments were valuable for improving our manuscript and guiding our research. We have studied the comments carefully and have made revisions accordingly. We hope that the revisions meet with your approval. The revised portions are marked in red font in the manuscript. Detailed point-by-point descriptions of the revisions are provided below.

General comments:

There is a similar article by the same authors in Remote Sensing (DOI: 10.3390/rs12244098). The current submission is titled ‘Spatial-temporal distribution of the freeze-thaw cycle of the largest lake (Qinghai Lake) of China based on the machine learning and MODIS from 2000 to 2020’ (I will call it A), while the previous paper it titled ‘Lake phenology of freeze-thaw cycles using random forest: a case study of Qinghai Lake’ (I will call it B). I don’t see any differences between the two titles, but anyway, I read the previous paper to see what might be different in this new one. Note that this new submission does not cite the previous one; but why did you publish a paper that you are not citing in your next similar paper? Usually we publish an article and then we cite this previous article to show where we paused our research, which is now advancing through the new article.

Anyway, the study aims are the following;

A: This work focuses on the ice phenology of Qinghai Lake based on MODIS images, Joint Research Center’s Global Surface Water (JRC GSW), and ERA5 dataset using the RF algorithm in the GEE platform.

B: The objective of the study was establishing or characterizing the lake phenology of Qinghai Lake based on MODIS remote sensing data using the RF algorithm in the GEE platform. What am I missing?

From the materials and methods I understand the following;

A: MODIS satellite images of the Qinghai Lake from 2000 to 2020 were acquired. An RF algorithm was trained and validated to analyze these images and discriminate ice and water areas across the lake from 2000 to 2020. The validated model was then used to actually predict the freeze-thaw cycles in the lake from 2000 to 2020 after removing clouds and filling the cloud-induced gaps.

  1. MODIS satellite images of the Qinghai Lake from 2000 to 2018 were acquired. An RF algorithm was trained and validated to analyze these images and discriminate ice and water areas across the lake from 2000 to 2018. The validated model was then used to actually predict the freeze-thaw cycles in the lake from 2000 to 2018 after removing clouds and filling the cloud-induced gaps, and the results are compared with other algorithms/models.

A different approach is applied between A and B to validate the RF algorithm. A: Sentinel-2 reflectance + JRC GSW + ERA5 data + cloud removal/gap-filling; B: MODIS reflectance imagery + cloud removal/gap-filling.

So, after reading both papers, I can’t see why a new paper is necessary to describe the freeze-thaw cycles in the Qinghai Lake for pretty much the same period. I suggest that the authors should decide on the focus of the paper. Will the focus be to compare between different RF optimization schemes, different training and validation options using the Qinghai Lake as an example? Then re-write this paper as a follow-up comparison with the previous paper, apply much detail on what this new optimized approach included and what parameters were additionally used or adjusted to increase testing accuracy. Will the focus be on describing the spatiotemporal variation of the freeze-thaw cycles in the lake? Then re-write the paper without all the technical detail and focus on clearly describing the freeze-thaw cycles. But wasn’t this the aim of the previous paper? Anyway, from Figures 8 and 9, it is not clear when freezing and thawing occurs. The new information that I can see is section 4.5, which however is actually existing information (it is already there in the previous paper) presented in a different style (instead of freeze-up start and end, presented as ice duration etc.).

Response : Thanks for the comments, we have added and modified the sentences “The previous research analyzed the water index, ice threshold method, and RF algorithm, and obtained the lake ice phenology [19]. This work is a continuation of the previous research. It focuses on the spatial-temporal distribution of ice phenology of Qinghai Lake based on MODIS images, Joint Research Center’s Global Surface Water (JRC GSW), and ERA5 dataset using the RF algorithm in the GEE platform and applys the cloud gap-filling method to reconstruct the gap area” on lines 74-79 to explain this work is a continuation of the previous research and differences.

A different approach is applied between A and B to validate the RF algorithm. A: Sentinel-2 reflectance + JRC GSW + ERA5 data + cloud removal/gap-filling; B: MODIS reflectance imagery + cloud processes (cloud threshold).

The sentence “Based on the comparative analysis of water index, ice threshold value method and RF algorithm in the previous research [19], the latter is very robust and can handle complex data, the main advantage is the bagging to avoid overfitting” on lines 151-153 was added.

The sentence “The critical parameters of RF algorithm are numberofTrees, variablesPerSplit, min-LeafPoplation, bagFraction, outofBagMode, and seed variables. The number of trees de-pends on the training data, the variablesPerSplit is 0.66, other variables could set default” on lines 159-162 was added.

The sentence “The grid search was performed, the previous article has set parameter in detail [19] ” on lines 189-190 was added.

Above added sentences is the continuation of the previous research, using RF optimization schemes, different training and validation options using the Qinghai Lake as an example. The pixel scale of lake ice/water transition was extracted and the cloud gap-filling method was applied to reconstruct the gap area.

Other comments

Point 1: Images and tables should be self-explained (i.e. read and understood without help from the main text).

Response 1: Thanks for the comments, we have modified according to the specific comments.

The sentence “the number of classes is four, samples dataset include seven bands” was added in figure 2 on line 201.

The sentence “(ai stands for the total number of actual class i pixels, bj stands for the total number of predicted class j pixels, and Nij stands for the total number of actual class i and predicted class j pixels, N stands for the total number of pixels, q stands for the number of classes)” was added in table 1 on lines 237-240.

The sentence “The dotted line above is the maximum, the dotted line below is the maximum, the solid line in the middle is the average” was added in figure 4 on lines 334-335.

The sentence “Bold font is the number of pixels of predicted values that are consistent with the actual values” was added in table 2 on lines 346-347.

The sentence “Bold font is the number of pixels of predicted values that are consistent with the actual values” was added in table 3 on lines 348-349.

The sentence “a-d is the date” was added in figure 5 on line 372.

The sentence “a-f is the date” was added in figure 6 on line 393.

The sentence “a-f is the date.” was added in figure 7 on line 396.

The sentence “Bold font is the number of pixels of predicted values that are consistent with the actual values” was added in table 4 on lines 382-383.

The sentence “Bold font is the number of pixels of predicted values that are consistent with the actual values” was added in table 5 on lines 414-415.

The sentence “the y axis scale is 1 (dense) or 2 (sparse) days; b. duration days and variation trend, f-DFU represented the fitted DFU, c. precipitation; d.2m air temperature. f-DFU stands for the fitted line of DFU; f-DBU stands for the fitted line of DBU; f-DFI stands for the fitted line of DFI; f-DI stands for the fitted line of DI; f-monthly stands for the fitted line of monthly value; f-yearly stands for the fitted line of yearly value” was added in figure 11 on lines 546-550.

Point 2: Throughout the text, replace ‘Figure XX showed’ by ‘Figure XX shows’. The same goes for the tables.

Response 2: Thanks for the comments, we have carefully modified the context, the phrases ‘Figure XX showed’ were replaced by ‘Figure XX shows’, the phrases ‘Table XX showed’ were replaced by ‘Table XX shows’.

The phrases was modified as follows:

  • Line 283: Figure 5(2), 5(5) shows.
  • line 292: Figure 3(a) and 3(b) shows.
  • line 307: Figure 3(c) and 3(d) shows.
  • line 354: Figure 5(1) and 5(3) shows.
  • line 355: Figure 5(2) and 5(4) shows.
  • line 397: Figure 6(1) and (2) shows.
  • line 403: Figure 7 shows.
  • line 404: Figure 7 (e) shows.
  • Line 471: Figure 10 shows.
  • line 491: Figure 11a shows.
  • line 337: Table 2 shows.
  • line 342: Table 3 shows.
  • line 378: Table 4 shows.
  • line 100: Table 5 shows.

Point 3: Figure 11. The abbreviations DFU, DFI, DBU, DI and DFW should be explained in the caption. What is f-yearly, f-monthly, f-DBU etc.? in Figure 11a, what kind of y axis scale is this? 10-20-Dec-10-20-Jan etc. Add another column to the left and write the month, and change this column to 10-20-30-10-20-30 … In Figure 11b you say ‘duration days/days’, is this correct? Do you mean ‘duration days (days)’? Then there is no need for ‘(days)’. Precipitation/m should be obviously written as ‘Precipitation (m)’. Does the pink color represent yearly precipitation or monthly? Similarly, the blue color is probably daily precipitation? Since you have values between months it can’t be monthly.

Response 3: Thanks for the comments, we have added the more details, the abbreviations DFU, DFI, DBU, DI and DFW was explained on the lines 268-273 in the section 3.4. The ‘f-yearly’, ‘f-monthly’, ‘f-DBU’ was explained on the lines 548-550 in the caption, that is “f-DFU stands for the fitted line of DFU; f-DBU stands for the fitted line of DBU; f-DFI stands for the fitted line of DFI; f-DI stands for the fitted line of DI; f-monthly stands for the fitted line of monthly value; f-yearly stands for the fitted line of yearly value”. The “y axis scale is 1 (dense) or 2 (sparse) days” was added in Figure 11a to explain what kind of y axis scale. The column was added on the subfigure (a), and the column was modified as the 10, 20, 1/30, …. The “duration days (days)” was modified as the “duration days”. The “Precipitation/m” was maintained, keep it consistent with the other formats. The pink color represents the yearly precipitation, and the blue color represents the monthly precipitation, since the x axis is the year.

Point 4: Results and Discussion. But actually, there is no discussion. You should show the results and then discuss them based on previous literature to try to find the factors that produced these trends. You can compare your results for this lake with possible results from other lakes in China to identify common trends etc. If the article’s focus is the algorithms and their optimization, you could compare with other studies around the world that apply other algorithms and discuss on the predictive accuracy etc. But now you only show us the results. It would be better to divide Results and Discussion to two different sections.

Response 4: Thanks for the comments, we have modified according to the specific comments.

Old version section 4.5 was the discussion of the section 4 results and discussion, now, we have supplemented the detailed discussion in section 5 discussion.

The discussion information was supplemented “The FUS and FUE showed a tendency to delay, while the BUS and BUE showed a tenden-cy to advance. The most former studies of lake ice had also showed a global warming tendency, and the latter was more intense change than the former [56,59-62]. ” on lines 495-498.

The discussion information was supplemented “It is consistent with the results that the ice duration shortened 17.5 days/decade for Cana-da’s lakes during 1985-2004 [3]. ” on lines 507-508.

The discussion information was supplemented “ The temperature began to increase in 2004/2005, 2010/2011 ice season year, the precipita-tion started to increase in 2002/2003, 2008/2009 ice season year. The relationship between the Polish Lowland lakes ice and winter air temperature was analyzed, the ice cover has been getting shorter under the climatic changes [59].” on lines 519-522.

The discussion information was supplemented “ In addition, the lake size, snowfall (area and thickness) and the ice thickness displayed the significant role in regulating the ice phenology [11]. ” on lines 532-534.

Point 5: Line 394. Are you sure about the Julian Days? If you convert 1/1/2001 to the Julian Day here https://www.aavso.org/jd-calculator it does not produce the values you mention. See here also https://sciencing.com/calculate-julian-date-6465290.html.

Response 5: Thanks for the comments, we have referred your suggestions about the Julian day, and supplemented the detail “The general calculation method of Julian date referred to the literature [10,26,56], i.e., when this day occurs within the year (day of year, DOY, with 1 January as the reference), the consecutive date of the next year is added.” On lines 258-260 to explain the Julian Days.

Point 6: Line 410. What is the difference between B: Figure 7 and Table A3, and A: Figures 8 and 9? I mean, they show the freeze-up and break-up dates but are these dates different between A and B? Did the optimized RF algorithm applied in B produce different freeze-thaw periods than the RF algorithm in A? From the figures this is not clearly visible.

Response 6: Thanks for the comments, we have added the information “The previous work mainly studied the lake ice as a whole, while this work focused on each pixel of the lake for detailed research” on lines 417-418, and “pixel-by-pixel” on lines 443 and 465 to explain the difference between A and B, B: Figure 7 and Table A3, and A: Figures 8 and 9.

Point 7: To conclude, I think that the focus of this paper should be clearly re-stated, whether it would be the optimization process itself or the description of the spatiotemporal variation of freeze-thawing processes in the lake (but the freeze-thawing processes have been described in the previous paper, haven’t they?). The manuscript should thus be re-written based on the selected focus as I also stated above.

Response 7: Thanks for the comments, we have modified and added the information in the context to re-state clearly the spatiotemporal variation of freeze-thawing processes in the lake and this work is a continuous work: the lake was studied as a whole in the previous work; the lake was composed by many pixels, pixel-by-pixel level was studied in this work.

Reviewer 3 Report

The current study focuses on the reconstruction of the spatio-temporal distribution of the water-ice structure in the Qinghai Lake in China for the period 2000-2020. Please consider addressing the suggestions and comments below:

1) It is not clear in which variables exactly the reconstruction took place. Did the authors use the machine learning algorithms on the water-ice structure of the entire period from 2000-2020 or did they used a smaller calibration and confirmation period to train their model. Also, can you please indicate which figure shows the comparison between the observed spatio-temporal variables and the simulated ones.

2) It is not clear how did all the rest variables at the Lake (such as precipitation, temperature etc.) contributed in the analysis. Please consider explaining in steps the methodology and how each variable shown in the analysis helped with the reconstruction.

3) In the Abstract and the rest of text, the authors mention that the increase in temperature resulted in an increase in precipitation after two years. However, I could not find how this statement is supported by the analysis. Please consider performing cross-correlation methods to estimate this lag between temperature and precipitation.

4) Also, please include some references from the literature that can support this statement. The climate dynamics is very complex and usually involves most of the hydrometeorological variables, and an increase in temperature does not necessarily result in an increase in precipitation. In fact, an increase in temperature may result in a decrease in precipitation due to the dew-point increase. Therefore, I think that the humidity should be included in the analysis to investigate how these two variables (temperature and precipitation) interact with each other (e.g., Koutsoyiannis, 2012; Ali et al., 2018).

5) The variability of temperature, precipitation, wind and other processes that affect the water-ice distribution structure may be also explained by scaling effects that have been identified from global analyses. For example, it has been shown that the above processes exhibit a long-term persistence behaviour (or else called Hurst-Kolmogorov dynamics) that can explain the strong observed variability (e.g., Kantelhardt et al., 2003; 2006; Dimitriadis and Koutsoyiannis, 2018; Koutsoyiannis et al., 2018). Please consider discussing this alternative explanation for the observed variability.

References

Ali, H.; Fowler, H.J.; and Mishra, V. Global observational evidence of strong linkage between dew point temperature and precipitation extremes.Geophys. Res. Lett.,45, 12320–12330, 2018.

Dimitriadis, P., and D. Koutsoyiannis, Stochastic synthesis approximating any process dependence and distribution, Stochastic Environmental Research & Risk Assessment, 32 (6), 1493–1515, doi:10.1007/s00477-018-1540-2, 2018.

Kantelhardt, J.W., D. Rybski, S.A. Zschiegner, P. Braun, E. Koscielny-Bunde, V. Livina, S. Havlin, A. Bunde, Multifractality of river runoff and precipitation: comparison of fluctuation analysis and wavelet methods,Physica A330, 240, 2003.

Kantelhardt, J. W., E. Koscielny-Bunde, D. Rybski, P. Braun, A. Bunde, and S. Havlin, Long-term persistence and multifractality of pre-cipitation and river runoff records, J. Geophys. Res.,111, D01106,doi:10.1029/2005JD005881, 2006.

Koutsoyiannis, D., Clausius-Clapeyron equation and saturation vapour pressure: simple theory reconciled with practice, European Journal of Physics, 33 (2), 295–305, doi:10.1088/0143-0807/33/2/295, 2012.

Koutsoyiannis, D., P. Dimitriadis, F. Lombardo, and S. Stevens, From fractals to stochastics: Seeking theoretical consistency in analysis of geophysical data, Advances in Nonlinear Geosciences, edited by A.A. Tsonis, 237–278, doi:10.1007/978-3-319-58895-7_14, Springer, 2018.

Author Response

Dear Editors and Reviewers,

Thank you for your letter and comments concerning our manuscript entitled "Spatial-temporal distribution of the freeze-thaw cycle of the largest lake (Qinghai Lake) of China based on the machine learning and MODIS from 2000 to 2020 " (remotesensing-1111572). The comments were valuable for improving our manuscript and guiding our research. We have studied the comments carefully and have made revisions accordingly. We hope that the revisions meet with your approval. The revised portions are marked in red font in the manuscript. Detailed point-by-point descriptions of the revisions are provided below.

Point 1: It is not clear in which variables exactly the reconstruction took place. Did the authors use the machine learning algorithms on the water-ice structure of the entire period from 2000-2020 or did they used a smaller calibration and confirmation period to train their model. Also, can you please indicate which figure shows the comparison between the observed spatio-temporal variables and the simulated ones.

Response 1: Thanks for the comments, we have modified and added the context according to the specific comments to explain clearly.

The sentence “The previous research analyzed the water index, ice threshold method, and RF algo-rithm, and obtained the lake ice phenology [19]. This work is a continuation of the previ-ous research. It focuses on the spatial-temporal distribution of ice phenology of Qinghai Lake based on MODIS images (2000-2020), Joint Research Center’s Global Surface Water (JRC GSW), and ERA5 dataset using the RF algorithm in the GEE platform and applys the cloud gap-filling method to reconstruct the gap area.” on lines 74-79 was added and modified to explain which variables exactly the reconstruction took place.

The sentence “84 images were selected over the time of 24th Feb. 2000 to 31st Dec. 2020” on lines 171-172, and “The date range of MODIS/Terra data was from 24th, Feb. 2000 to 31st, Jul. 2020. The date range of MODIS/Aqua data was from 4th, Jul. 2002 to 31st, Jul. 2020” on lines 109-113 to explain the entire period.

The Figures 6 and 7 are the cloud gap-filling procedure for entire lake, Figures 8 and 9 are the Julian day of ice freeze-up and break-up pixel-by-pixel during 2000-2020 using RF algorithm, the simulated model was not used.

Point 2: It is not clear how did all the rest variables at the Lake (such as precipitation, temperature etc.) contributed in the analysis. Please consider explaining in steps the methodology and how each variable shown in the analysis helped with the reconstruction.

Response 2: Thanks for the comments, we have added and modified the context.

The sentence “Based on the comparative analysis of water index, ice threshold value method and RF algorithm in the previous research [19], the latter is very robust and can handle complex data, the main advantage is the bagging to avoid overfitting.” on lines 151-153.

The sentence “The critical parameters of RF algorithm are the number of trees, variablesPerSplit, min-LeafPoplation, bagFraction, outofBagMode, and seed variables. The number of trees de-pends on the training data, the variablesPerSplit is 0.66, other variables could set default.” on lines 159-162.

The sentence “the grid search was performed, the previous article has set parameter in detail [19].” on lines 189-190.

The sentence “ai stands for the total number of actual class i pixels, bj stands for the total number of predicted class j pixels, and Nij stands for the total number of actual class i and predicted class j pixels” on lines 233-235.

and the Cloud gap-filling procedure in Figures 6 and 7 to explain in steps the methodology and how each variable shown in the analysis helped with the reconstruction.

Point 3: In the Abstract and the rest of text, the authors mention that the increase in temperature resulted in an increase in precipitation after two years. However, I could not find how this statement is supported by the analysis. Please consider performing cross-correlation methods to estimate this lag between temperature and precipitation.

Response 3: Thanks for the comments, we have modified according to specific comments.

The sentence “The temperature began to increase in 2004/2005, 2010/2011 ice season year, the precipitation started to increase in 2002/2003, 2008/2009 ice season year. The relationship between the Polish Lowland lakes ice and winter air temperature was analyzed, the ice cover has been getting shorter under the climatic changes [59]” was added on lines 519-522 to explain the conclusion “the increase in the temperature, resulting in an increase of precipitation after two years”.

In addition, the ‘f-yearly’, ‘f-monthly’, ‘f-DBU’ was explained on the lines 547-550 in the caption, that is “f-DFU stands for the fitted line of DFU; f-DBU stands for the fitted line of DBU; f-DFI stands for the fitted line of DFI; f-DI stands for the fitted line of DI; f-monthly stands for the fitted line of monthly value; f-yearly stands for the fitted line of yearly value”. The “y axis scale is 1 (dense) or 2 (sparse) days” was added in Figure 11a to explain what kind of y axis scale. The column was added on the subfigure (a), and the column was modified as the 10, 20, 1/30, …. The “duration days (days)” was modified.

Point 4: Also, please include some references from the literature that can support this statement. The climate dynamics is very complex and usually involves most of the hydrometeorological variables, and an increase in temperature does not necessarily result in an increase in precipitation. In fact, an increase in temperature may result in a decrease in precipitation due to the dew-point increase. Therefore, I think that the humidity should be included in the analysis to investigate how these two variables (temperature and precipitation) interact with each other (e.g., Koutsoyiannis, 2012; Ali et al., 2018).

Response 4: Thanks for the comments, we have added and modified the context according to the specific comments.

The sentence “In addition, the lake size, snowfall (area and thickness) and the ice thickness displayed the significant role in regulating the ice phenology [11], humidity is a very sensitive variable that responds to changes in lake phenology and its surrounding environment [64,65].”, the some references were cited on lines 532-535.

Point 5: The variability of temperature, precipitation, wind and other processes that affect the water-ice distribution structure may be also explained by scaling effects that have been identified from global analyses. For example, it has been shown that the above processes exhibit a long-term persistence behaviour (or else called Hurst-Kolmogorov dynamics) that can explain the strong observed variability (e.g., Kantelhardt et al., 2003; 2006; Dimitriadis and Koutsoyiannis, 2018; Koutsoyiannis et al., 2018). Please consider discussing this alternative explanation for the observed variability.

Response 5: Thanks for the comments, we have added and modified the context according to the specific comments.

The sentence “The theoretical consistency in analysis of geophysical data from Fractals to Stochastics method were also critical [66-69].” on lines 535-537.

The sentence “It is consistent with the results that the ice duration shortened 17.5 days/decade for Canada’s lakes during 1985-2004 [3].” on lines 507-508.

The sentence “The relationship between the Polish Lowland lakes ice and winter air temperature was analyzed, the ice cover has been getting shorter under the climatic changes [59].” on lines 520-522.

The some references were cited.

Reviewer 4 Report

The manuscript "Spatial-temporal distribution of the freeze-thaw cycle of the largest lake (Qinghai Lake) of China based on the machine learning and MODIS from 2000 to 2020" presents a case study applying machine learning (random forest algorithm) based on satellite imagery (MODIS) to analysis freezing cycles.

The manuscript is well organized easy to read. The authors described the methods and results adequately. However, there are few points that I think it is important to address/clarify before accepting this manuscript for Remote Sensing.

Did the authors perform some correction/calibration in ERA5 data? One concern about the precipitation provided by ERA5 is the drizzle effects. In the annual timescale, the drizzle may not affect the results since the yearly rainfall will be mostly consistent with in site observation, but the daily timescape will certainly be affected.

Although the authors described the results extensively, the discussion needs attention. Not only because there are few citations, but the authors could highlight previous research on this topic. The authors just ignore them, and maybe because of this, the section "Results and Discussion" is basically "Results."

There are few comments below for specific lines.

Lines 14-16: Split into two sentences.

Lines 66-67: Inform the latency of images availability;

Lines 108-110: Standardize the satellite description (250 meters our 250m)

Figure 4. Please inform on the caption what the horizontal lines mean;

Fix color bar of figure 9;

Figure 10: Recommend use lines and points and precipitation represented with bars; In addition, if keeping lines, differ the lines to facilitate the interpretation of this figure if it is printed in gray colors.

Standardize the units (with or without space between the number and units).

Author Response

Dear Editors and Reviewers,

Thank you for your letter and comments concerning our manuscript entitled "Spatial-temporal distribution of the freeze-thaw cycle of the largest lake (Qinghai Lake) of China based on the machine learning and MODIS from 2000 to 2020 " (remotesensing-1111572). The comments were valuable for improving our manuscript and guiding our research. We have studied the comments carefully and have made revisions accordingly. We hope that the revisions meet with your approval. The revised portions are marked in red font in the manuscript. Detailed point-by-point descriptions of the revisions are provided below.

Point 1: Did the authors perform some correction/calibration in ERA5 data? One concern about the precipitation provided by ERA5 is the drizzle effects. In the annual timescale, the drizzle may not affect the results since the yearly rainfall will be mostly consistent with in site observation, but the daily timescape will certainly be affected.

Response 1: Thanks for the comments, we have carefully check the ERA5 data, the hourly precipitation provided by ERA5 is the drizzle effects, the daily precipitation provided by ERA5 is not the drizzle effects.

The more detail has been added “Monthly total precipitation is the sum of all daily precipitation within the month, yearly total precipitation is the sum of all daily precipitation within the year; monthly tempera-ture is the mean value of all daily temperature within the month, yearly temperature is the mean value of all daily temperature within the year.” on lines 141-145.

Point 2: Although the authors described the results extensively, the discussion needs attention. Not only because there are few citations, but the authors could highlight previous research on this topic. The authors just ignore them, and maybe because of this, the section "Results and Discussion" is basically "Results."

Response 2: Thanks for the comments, we have added and modified the detailed discussion.

The sentence “The previous work mainly studied the lake ice as a whole, while this work focused on each pixel of the lake for detailed research.” on lines 416-417 has been added in section 4.4.

The sentence “The FUS and FUE showed a tendency to delay, while the BUS and BUE showed a tenden-cy to advance. The most former studies of lake ice had also showed a global warming tendency, and the latter was more intense change than the former [56,59-62]” on lines 493-496 has been added.

The sentence “It is consistent with the results that the ice duration shortened 17.5 days/decade for Cana-da’s lakes during 1985-2004 [3].” on lines 505-506 has been added.

The sentence “The temperature began to increase in 2004/2005, 2010/2011 ice season year, the precipitation started to increase in 2002/2003, 2008/2009 ice season year. The relationship between the Polish Lowland lakes ice and winter air temperature was analyzed, the ice cover has been getting shorter under the climatic changes [59].” on lines 517-520 has been added.

The sentence “In addition, the lake size, snowfall (area and thickness) and the ice thickness displayed the significant role in regulating the ice phenology [11], humidity is a very sensitive variable that responds to changes in lake phenology and its surrounding environment [64,65]. The theoretical consistency in analysis of geophysical data from Fractals to Stochastics method were also critical [66-69] .” on lines 530-535 has been added.

The sentence “the y axis scale is 1 (dense) or 2 (sparse) days; b. duration days and variation trend, f-DFU represented the fitted DFU, c. precipitation; d.2m air temperature, f-DFU stands for the fitted line of DFU; f-DBU stands for the fitted line of DBU; f-DFI stands for the fitted line of DFI; f-DI stands for the fitted line of DI; f-monthly stands for the fitted line of month-ly value; f-yearly stands for the fitted line of yearly value). ” on lines 544-548 has been added in caption.

Point 3: Lines 14-16: Split into two sentences.

Response 3: Thanks for the comments, we have modified the sentence to two sentences.

The sentences “Based on the MODIS imagery, the spatio-temporal dynamics of the ice phenology of Qinghai Lake was analyzed using machine learning during the 2000/2001 to 2019/2020 ice season” on lines 14-16.

Point 4: Lines 66-67: Inform the latency of images availability

Response 4: Thanks for the comments, we have added the sentence “The latency of images availability is 21 days.” on line 73 to inform the latency of images availability.

Point 5: Lines 108-110: Standardize the satellite description (250 meters our 250m)

Response 5: Thanks for the comments, we have modified the sentence “visible and NIR (10 m), red edge, and SWIR (20 m), and atmospheric bands (60 m)” on lines 119-120 to Standardize the satellite description.

Point 6: Figure 4. Please inform on the caption what the horizontal lines mean

Response 6: Thanks for the comments, we have added the sentence “The dotted line above is the maximum, the dotted line below is the maximum, the solid line in the middle is the av-erage” on lines 334-335 to explain on the caption what the horizontal lines mean.

Point 7: Fix color bar of figure 9

Response 7: Thanks for the comments, the color bar of Figure 9 (a) is similar to the color bar of Figure 8 (a). The smaller value (earlier freeze-up or break-up date) corresponds to a bluer color, and a larger value (later freeze-up or break-up date) corresponds to a redder color.

Point 8: Figure 10: Recommend use lines and points and precipitation represented with bars; In addition, if keeping lines, differ the lines to facilitate the interpretation of this figure if it is printed in gray colors.

Response 8: Thanks for the comments, we have modified the Figure 10 according to the specific comments, the Figure 10 is as follow:

Point 9: Standardize the units (with or without space between the number and units).

Response 9: Thanks for the comments, we have carefully checked the space between the number and units to standardize the units, retained the space between the number and units.

The “1.9 °C” on line 92;

The “-11.4 ℃ and 12.5 ℃” on line 93.

The “500 m” on line 103.

The “2 m” on line 140.

The “2 m” on line 525.

The “2 m” on line 546.

Round 2

Reviewer 3 Report

All the review comments were addressed and discussed in the revision.

Author Response

Dear Editors and Reviewers,

Thank you for your letter and comments concerning our manuscript entitled "Spatial-temporal distribution of the freeze-thaw cycle of the largest lake (Qinghai Lake) of China based on the machine learning and MODIS from 2000 to 2020 " (remotesensing-1111572). The comments were valuable for improving our manuscript and guiding our research. We have studied the comments carefully and have made revisions accordingly. We hope that the revisions meet with your approval. The revised portions are marked in red font in the manuscript. Detailed point-by-point descriptions of the revisions are provided in “Reply to Academic Editor”.

Reviewer 4 Report

The authors properly addressed all of my comments and at this point, I do recommend the acceptance of the manuscript.

Author Response

(The authors gave the same response as above.)
